# Cultural framing of giftedness in recent US fictional texts

**Daniel Patrick Balestrini** *, **Heidrun Stoeger**

Chair of School Research, School Development, and Evaluation, Institute of Education Science, Faculty of Human Sciences, University of Regensburg, Regensburg, Bavaria, Germany

* daniel-patrick.balestrini@ur.de

## Abstract

A perennial topic of research on giftedness has been individuals' perceptions of and attitudes towards giftedness, the gifted, and gifted education. Although giftedness is a culturally constructed concept, most examination of the term's meanings and implications has used reactive measures (i.e., surveys) to tap respondents' giftedness-related perceptions and attitudes within the context of formal education. To provide a better understanding of the cultural meanings associated with giftedness—the term's cultural framing—we investigated the depiction of giftedness within a professional cultural product removed from education, namely, a large corpus of US fictional texts. We examined patterns of word usage in the vicinity of the term *gift**, when used in the dictionary senses related to giftedness, in a large corpus of US fictional texts of recent decades, consisting of 485,179 text samples and 1,002,889,754 word tokens. Via inductive methods of quantitative text analysis, we explored themes occurring in the vicinity of *gift**; and with an existing lookup dictionary, we assessed deductively the overall emotional valance of the writing near *gift**. Our investigation revealed ways in which the literary exploration of giftedness coheres with and distinguishes itself from the outlooks on giftedness noted for survey-based research in education settings. In fictional texts, giftedness evinces special associations with humanities domains and beauty and, on balance, correlates positively with emotionally positive words.

**Data Availability Statement:** The US fiction data we used for our study are from the Corpus of Contemporary American English (COCA). References are provided for the COCA in the main manuscript. We used a version of the COCA licensed and downloaded in early 2020. We cannot share the corpus data due to license restrictions.

## Introduction

Informal giftedness-related outlooks have a big impact on individuals. They affect how gifted individuals are perceived in society, who is nominated for giftedness programs, and what educational experiences they have [1–3]. Giftedness-related outlooks also influence whether societies embrace the development of giftedness and talent or possibly foster a "cult of mediocrity" [4] (p.11) [5]. Despite knowing that giftedness-related outlooks affect individuals and societies, there is insufficient, contradictory knowledge about informal conceptions of giftedness [2, 6–8].

While hundreds of studies have tapped individuals' thoughts on giftedness, only a handful of studies has sought to understand the everyday meanings of giftedness by considering references to giftedness within cultural products. This, however, is important as giftedness is widely understood to reflect a culturally rooted concept [9, 10]. An especially promising approach to

However, users can license and then download the same COCA data via the site hosting the COCA. We provide a link to that site in the "Data Availability Statement" section of our S1 Technical Supplement. As of 18 February 2024, the COCA data were still available for licensing and downloading at that site. We created a repository with data sets used when conducting and reporting our study (https://www.doi.org/10.5283/epub.55543). We also created a revised version of the repository as we made slight changes to our analyses during the revision process (https://www.doi.org/10.5283/epub.58477). It contains all the data sets we used for calculating all values reported in our study with the exception of the text files and derivative data sets containing passages from the COCA. Currently, as of manuscript submission, both versions of the repository have restricted access. Editors and reviewers considering our manuscript can access the repository by clicking on the "request access" button. The repository is run by the University of Regensburg Library, and the responsible open access office knows we will be submitting our manuscript to PLOS ONE. When editors or reviewers reviewing our manuscript request access to our repository, access will be granted to them. We plan to make the repository completely open access immediately when our study has been published. In most cases, the analyses we describe and values we report in our manuscript and in the Technical Supplement in more detail can be verified using the data available in our repository. Once users have licensed and downloaded the COCA and extracted the relevant text files, as described in the "Corpus Ingestion and Description" section of our Technical Supplement, it will be possible to replicate our entire study from the ground up by following the instructions in the supplement.

**Funding:** The author(s) received no specific funding for this work.

**Competing interests:** The authors have declared that no competing interests exist.

understanding giftedness conceptions at the non-individual, cultural level lies in the investigation of cultural products (i.e., artifactual evidence of cultural outlooks and norms) [11, 12]. By systematically examining cultural products (e.g., social media, advertising, or fiction) created in circumstances unrelated to the investigation of giftedness, researchers can gain a better understanding of general cultural views on giftedness [13–15]. Yet, cultural products have rarely been examined regarding widespread notions of giftedness.

With our study, we sought to expand the knowledge base on the cultural underpinnings of giftedness by investigating cultural framings of giftedness in text-based cultural products, namely in fictional texts originally published in the United States in recent decades. We looked specifically for cultural-products-based answers to two of the most vexed questions within research on everyday conceptions of giftedness: the salient definitional characteristics of giftedness and the sentiments associated with giftedness.

## Survey investigations on outlooks on giftedness

Hundreds of studies have elicited respondents' views of and sentiments towards giftedness, the gifted, and gifted education with the help of surveys [16–19]. Results generally show that respondents associate giftedness with intellectual precocity. Beyond that hallmark, however, findings are inconsistent, with respondents noting various defining characteristics (e.g., creativity, motivation, and noncognitive personality traits) differing across studies. Similarly incoherent are findings relating to individuals' sentiments towards giftedness and related concepts [20, 21]. While some studies find respondents generally positive about giftedness constructs, most of them come to opposite conclusions, describing negative appraisals [1, 22, 23]. One aspect that seems to influence the view about giftedness is exposure to gifted education. Here the research literature suggests a clear relationship. The more exposure respondents have had to gifted education as trainees or teachers, for example, the more positive, albeit not necessarily more accurate, their thinking about giftedness tends to be [23, 24].

However, most of the survey investigations of outlooks on giftedness were conducted with a narrow range of populations with close connections to the topic of gifted education or giftedness. The groups in the focus of most research are limited to educational agents such as teachers and preservice teachers [8, 21, 23], administrators [25], school psychologists and counselors [26], gifted and nongifted students [27–31], parents and siblings [32, 33], and individuals seeking guidance about giftedness or gifted education services [7]. This focus is akin to attempting to understand widespread beliefs related to religiosity by surveying churchgoers. Examining populations with close connections to the topic of gifted education might be a good place to start, but the findings would be unrepresentative of society as a whole.

Two studies [22, 34] addressed this shortcoming by shifting the focus of the investigations of respondents' cognitions and emotions related to the *gifted* label from stakeholders in formal education to a representative sample of adults living in the United States and Germany respectively. In the US study [34], the 28-item Wiener Attitude Scale [35] was used to assess attitudes towards the gifted held by various groups within US society, including a subsample of $n = 130$ members of the "lay public," who comprised "a cross-section of people with no direct role involvement with the gifted" [34] (p.69). The survey indicated that the sentiments towards the gifted reported by this group were significantly more negative than those reported by the educational agents of gifted children.

Adding their survey items to a regular Germany-wide marketing survey, the author of the German study [22] surveyed a sample of adults living in Germany on their outlooks on gifted persons. The sample was characterized as being representative of adults aged 18–69 and living in Germany with respect to age, gender, and geographical distribution. In the study, a two-

thirds majority of respondents held negative stereotypes about the gifted with regard to socioe-motional traits.

Both studies [22, 34] provide insight into a more generalized level of informal views on gift-edness and avoid samples skewed towards 'insiders' in the education system. They nevertheless provide a weak proxy of the cultural framing of giftedness because cultural inferences based on the aggregation of individuals' views as assessed via questionnaires and interviews have been shown to provide weaker approximations of cultural outlooks than those based on artifactual data, that is, on cultural products [11, 12, 36, 37].

## Cultural-products investigations of outlooks on giftedness

The lack of fidelity in how individuals' views aggregate into cultural patterns lies in the meth-odological limitations of survey methods [38]. In particular, the phenomenon of reactivity among respondents [39, 40] manifests itself in the form of various measurement-induced biases. Reference group effects—in which survey responses reflect respondents' implicit self-comparisons against envisioned others or groups [41]—for example, are known to hamper the survey-based investigation of educationally relevant outlooks [42] including giftedness [7, 43, 44] and of questions involving cultural values more generally [45]. Reactivity also manifests in the form of nonresponse [46], which poses a crucial methodological shortcoming of survey-based inquiries into the cultural underpinnings of giftedness [7, 47].

Cultural products can provide a useful complementary method when investigating notions and sentiments about giftedness. The method circumvents the noted methodological problems by looking directly at the cultural substrate, which we term the *cultural framing of giftedness. Cultural products* are defined as tangible, public representations of culture of any sort that are amenable to study [37]. Cultural-products findings thus provide additional context for survey-based research findings.

Cultural products capture two sorts of cultural information [12]: (a) informal behavioral traces that provide insights into unscripted behavioral performances within social situations (e.g., a discussion of a topic on a social media forum) and (b) the products of professional pro-ducers of cultural products "who have the power to create and push a message" [12] (p. 697), such as books or commercials. Each type of cultural product provides a different sort of poten-tially valuable information for investigating cultural meanings and thus the cultural framing of giftedness.

**Informal versus professional cultural products.** Informal cultural products capture dynamic real-time or near-real-time responses to social situations, thus reflecting impromptu behaviors. Such cultural products may be similar to ephemeral non-written unscripted social interactions (e.g., a spontaneous interaction in public). Cultural products produced by skilled content creators—professional cultural products—tend to reflect the outlooks and meanings endorsed by institutions and individuals with more power in a given society or situation [12]. It has been postulated that professional creators of cultural products (e.g., novelists or advertis-ers) may be particularly adept at capturing subtle but pervasive cultural meanings that untrained producers of cultural products may overlook or not communicate as clearly [48]. Professional cultural products can therefore reflect more potent or succinct messaging and thus be particularly useful for investigating the cultural framing of ideas through cultural products.

**Cultural-products investigations of giftedness.** A comparatively small number of studies has examined cultural products for insights into the cultural framing of giftedness. Almost all studies have considered professional cultural products such as US television programming [49, 50]; US movies [50–52]; magazines and trade publications in Finland [53] and the United States

[54, 55]; websites in Estonia [56] and the United Kingdom [57]; newspapers in Finland [53], German-speaking Europe [58], the United Kingdom [59–61], and the United States [62, 63]; linguistically motivated corpora of written language in Estonia [56] and German-speaking Europe [13]; and Google books data for German-speaking Europe [13]. Three studies investigated giftedness-related informal cultural products, including fan content on a now defunct girls-focused UK website [57], informal content on a now defunct, but previously popular Estonian website [56], and giftedness-related discourse on the Reddit social-media platform [15].

These studies yield an incoherent picture, both regarding what giftedness is about and the cultural sentiment attached to the concept. Common themes are the association of cognitive precocity and socioemotional deficits [50, 51, 53]; giftedness being unhealthy for children [53, 54, 59–61]; interest in the feats and intellectual prowess of intellectually prodigious youths [64]; the gifted as representatives of the social elite [56]; and a salient association between giftedness and females [50] and between giftedness and the intellectual and mathematical [13].

While negative framings of giftedness appear dominant in the examined professional cultural products [51, 53, 54, 59–61], one recent study of informal cultural products reported a lack of evidence of a negative stance for a larger sample of giftedness-related discussions on the Reddit social-media platform [15], which is frequented by young adults in particular [65]. The Reddit study [15] considered the sentiment of the discussions, characterizing the overall sentiment as neutral. For a Reddit forum focusing on underachievement, the authors [15] had expected to observe a negative sentiment in line with existing findings on the topic. Finally, two diachronic investigations of sentiment surrounding giftedness and gifted children in professional cultural products observed a gradual amelioration of the sentiments associated with giftedness and gifted children in cultural products for recent decades [58, 64].

It is difficult to reconcile the disparate giftedness-related associations and sentiments being reported across the cultural-products investigations into giftedness. Three methodological shortcomings may help explain the lack of coherence among the cultural-products findings. First, most of the findings from the noted cultural-products studies lack generalizability due to the use of small samples and exploratory methods of data analysis. Seven cultural-products investigations into giftedness relate impressions from informal case studies dealing with only one or a few specific cultural products [50–52, 54, 59, 60, 64]. For example, one study [50] related details from two television series episodes, one novel series, and one website forum; another study [52] described circumstances in two films. Seven cultural-products investigations reported larger sample sizes [53, 55, 56, 58, 61–63]. However, all of these studies reported using exploratory methods of analysis that preclude making generalizations of any kind beyond the immediate samples. Only one study [15] considered a large sample of a specific type of informal cultural product, namely, social-media posts, and provided a detailed methodological description that would allow for replication and a reasonable level of confidence regarding the generalization of the results.

Second, none of the cultural-products investigations of giftedness systematically avoided capturing the perspectives of educational stakeholders. By failing to do so, it is likely that these studies are reflecting the special context of education and gifted education. Not counting case studies, four studies used *gifted education* or *gifted children* as their search terms [55, 61–63], which means that their findings invariably capture first and foremost the experiences of stakeholders in education and gifted education. While four studies used *giftedness* as their search term [13, 53, 56, 58], three of these [53, 56, 58] noted that education was a common topic among the collected documents. Moreover, one study within this group [53] used a Finnish trade publication aimed at educators as part of its sample, which ensured a focus on formal education despite using the more general search term.

While the Reddit study [15] investigating social-media posts overcomes the first noted limitation of earlier studies by systematically considering a large sample and therefore providing readily generalizable insights into the question of the sentiment attached to giftedness, the study's relevance for an investigation of widespread cultural outlooks on giftedness outside the educational context is also limited in the sense of the second noted shortcoming. The study considered discussions on Reddit focused on giftedness and gifted education, which were largely taking place among young adults and, in most cases, in forums (in so-called subreddits) linked closely to education, parenting, giftedness, or gifted education; of 12,105 samples, 11,771 samples (97.240%) were extracted from subreddits closely linked with education (i.e., from /r/teenagers, /r/teachers, /r/parenting, /r/gifted, /r/giftedconversations, and /r/after-gifted), while 334 samples (2.759%) were extracted from a generalized subreddit with no specific connection to education stakeholders (i.e., from /r/askreddit). In that sense, the findings are likely to reflect outlooks on giftedness of stakeholders with a close connection to education and gifted education.

Third, most of the cultural-products studies investigated articles appearing in newspapers or magazines [13, 14, 53–55, 58, 60–63]. While the quotidian perspective of news media is important for cultural-products investigations, the tendency of news media to stress and even contribute to problems and anxieties by highlighting bad news (i.e., negativity) and conflict [66] reflects a crucial limitation for the investigation of the cultural framing of giftedness. In so doing, news reporting is likely to amplify negative tropes and concerns related to giftedness in a society, potentially beyond their ambient levels across society in general.

## The present research

Wholly lacking among cultural-products investigations into the connotations and sentiment surrounding giftedness are investigations of literary texts such as novels and short stories. Investigations of fictional literary texts can avoid two of the methodological shortcomings of past research on cultural products. They have no particular affinity with formal education or the perspectives of parents and children; and they do not have the negativity bias found in news reporting. Furthermore, fictional literary texts provide extensive, nuanced insight into the norms and outlooks within cultures, as literature scholars have long noted. In a seminal investigation of the cultural relevance of American fiction [67], for example, the argument is made that a culture's literary works "offer powerful examples of the way a culture thinks about itself" [67] (p.xi). The investigation [67] identified the importance of literary texts as "agents of cultural formation" [67] (p.xvii). This understanding of the informational value of literary works for accessing the norms and outlooks that characterize societal culture (unfiltered through the lens of the psychological view of culture elicited via surveys of individuals' outlooks) dovetails with the assertions of the potential of professional cultural products to provide useful insights into culture at the level of societies [12, 48]. Moreover, recent digital humanities scholarship has arrived at a similar conclusion regarding the potential to better understand cultural meanings, subtexts, and sentiments via the systematic investigation of large numbers of literary texts through computational text analysis [68, 69]. Considering the potential of the computational text analysis of large numbers of professional cultural products in general and fictional literary texts in particular (i.e., fictional texts such as novels, short stories, and plays) to provide nuanced insights into the times, places, and social strata of their creation, an investigation of the depiction of giftedness in large samples of literary works can provide important insights into the cultural framing of giftedness and, currently, qualifies as a desideratum within the investigation of cultural constructions of giftedness.

To address the noted research desideratum, we investigated the cultural framing of giftedness in fictional literary texts published in recent decades in the United States. Building on earlier efforts to characterize the nature and sentiment of the cultural framing of giftedness via surveys and cultural-products investigations, we focused on two exploratory research questions:

1. Which notions of giftedness and the gifted are salient in fictional literary texts published in recent decades in the United States?

2. What is the overall sentiment valence when the terms *giftedness/gifted* are used in these texts?

## Materials and methods

### Overall study design

As a representative cultural product for this study, we selected an existing corpus of recent US fictional texts published since 1990 [70], which is part of the Corpus of Contemporary American English (COCA). The COCA was designed specifically for use as an object of investigation on matters relating to US language and culture [70]. We excerpted short passages surrounding all instances of the word *gift\** from the COCA fiction subcorpus for the period 1990–2017 and subjected these to three forms of quantitative text analysis in order to address the two research questions. To address Research Question 1 (RQ1), we implemented a keyword identification procedure to identify salient words in the vicinity of *gift\** and principal component analysis to identify clusters of words (themes) in the vicinity of *gift\**. To address Research Question 2 (RQ2), we implemented a sentiment analysis, using existing lookup dictionaries of positively and negatively connoted words [71] to gauge the positivity and negativity of the words found shortly before and after g*ift\** in our samples. The keyword analysis and the principal component analysis approach the language data inductively, while the sentimental analysis is deductive. All three analyses are descriptive and exploratory in nature and thus suited to a first-ever interrogation of a large sample of fictional text passages related to the concept of giftedness.

### Sample

We created our corpus from the fiction category (subcorpus) of the Corpus of Contemporary American English (the COCA) [70, 72, 73]. The COCA is a linguistically balanced corpus reflecting a range of spoken and written US English. The COCA qualifies as a balanced corpus in that it consists of equal proportions of language use sampled in different categories, making it possible for researchers to investigate language use in broad categories.

The COCA contains samples of English texts published since 1990 in the United States in the categories 'spoken,' fiction (including plays), magazines, newspapers, and academic journals [70, 72, 73], and additional data have been added recently for television and movies, blogs, and websites for the same period [74]. As of March 2020, the COCA consisted of 485,179 texts and 1,002,889,754 word tokens. The COCA thus systematically assembles a large sample of professional cultural products as defined above [12]. From that perspective, the COCA texts are tangible, public representations of US culture that provide artifactual evidence of the cultural norms that characterize and define US society and help determine individuals' cognitions, emotions, and behaviors.

For the period 1990–2019, the fiction subcorpus comprises 25,992 texts with 119,505,305 words and includes equal portions of "short stories and plays from literary magazines, children's magazines, popular magazines, first chapters of first edition books [. . .], and fan fiction"

[73]. Due to licensing restrictions in place in early 2020 when we downloaded the COCA, our data are limited to the period 1990–2017. Moreover, to respect US fair use law, the download-able version of the COCA systematically removes 5% of the original texts by removing 10 words every 200 words [75]. After downloading the entire COCA, we created our local corpus of US fictional texts in the following steps. See the Technical Supplement (S1 File), for additional details.

First, we extracted the 28 text files containing the fiction subcorpus for the period 1990–2017. The files together consist of 24,887 sampled texts, with an average of 888.82 texts per year ($SD$ = 258.19). The 1990–2017 fiction subcorpus consists of 108,225,089 word tokens, with an average of 3,865,181.75 word tokens per year ($SD$ = 172,553.38). On average, each sampled text consists of 4,349 word tokens ($SD$ = 6,290). The very large standard deviation reflects a long right tail in the distribution of the word counts of the sampled texts (Fig 1).

Second, we created two derivative corpora from the COCA fiction subcorpus for our subsequent investigation, a target corpus and a reference corpus. Our target corpus consisted of an

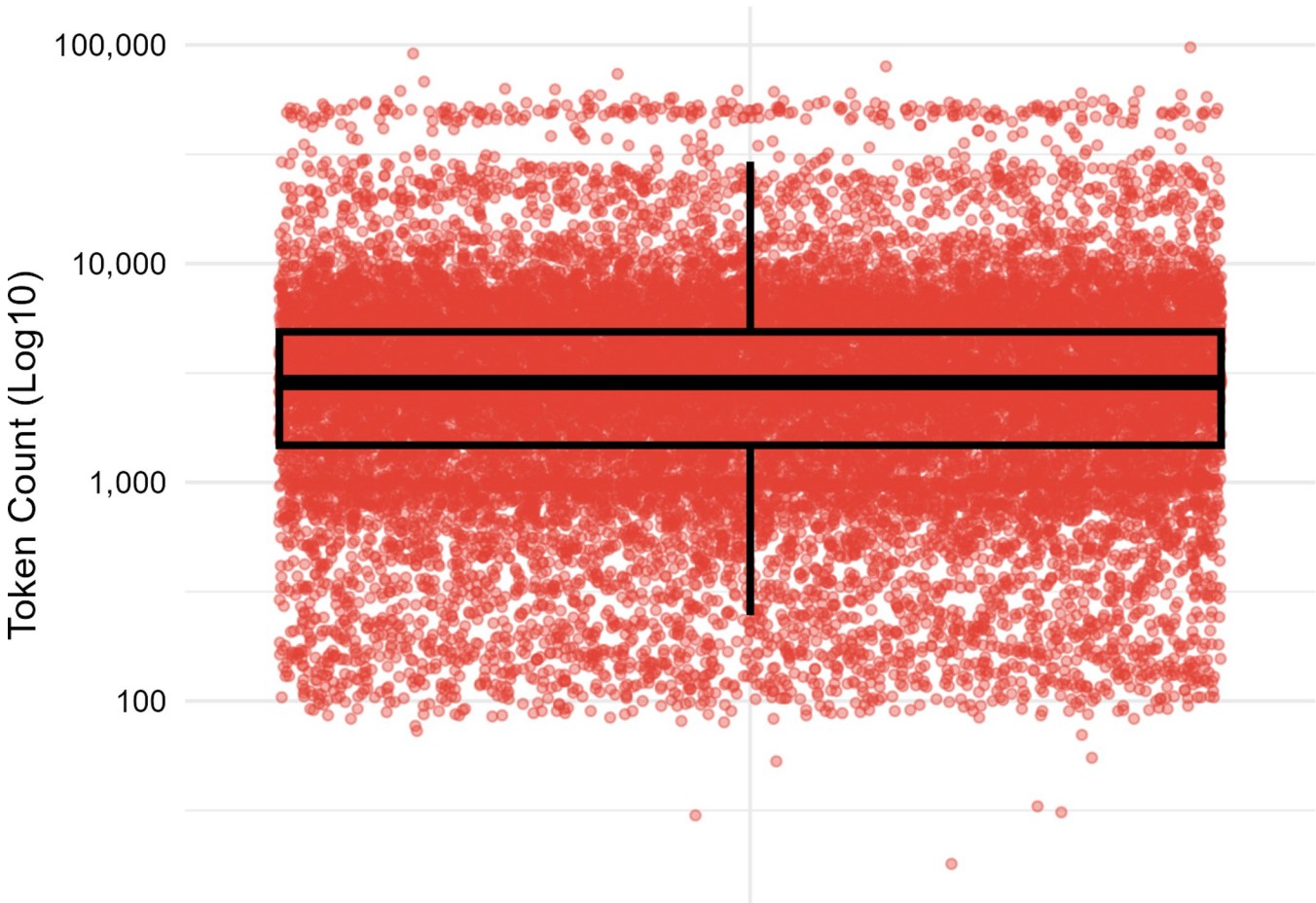

**Fig 1. Word count distribution among the sampled texts in the Corpus of Contemporary American English (COCA) fiction subcorpus.** Box plot with "jittered" circles [76] to show the distribution of text lengths. The red circles represent all text lengths in the COCA fiction subcorpus, including all outliers, randomly distributed from left to right (i.e., 'jittered') to better illustrate their distribution along the y-axis. The middle horizontal line in the box indicates the median document length (i.e., 2,860 word tokens). The lower horizontal line (lower hinge) demarcates the top of the lowest quartile of text lengths (i.e., 1,478 word tokens), into which 25% of the text lengths fall. The upper horizontal line (upper hinge) demarcates the top of the third quartile of text lengths (i.e., 4,879 tokens), below which 75% of the text lengths fall. The black vertical line extending below and above the box indicates the range of all values falling within 1.5 times the interquartile range indicated by the box.

exhaustive sample of short windows of text (i.e., passages) centered on *gift**. Our reference corpus consisted of an exhaustive sample of short windows of text centered on *only*. With the *gift** corpus (i.e., the target corpus), we collected all instances in which the target concept of giftedness, operationalized with the word *gift**, occurs in the COCA fiction subcorpus. The *only* corpus (i.e., the reference corpus) serves as a baseline for comparisons of language use in the *gift** corpus with language use in the COCA fiction subcorpus in general. As *only* is a grammar word, it is reasonable to assume that the word will occur unsystematically across the entire COCA fiction subcorpus, thereby reflecting word usage in general as it occurs throughout the entire COCA fiction subcorpus. The *only* corpus thus provides a reference or comparison corpus [14, 77] in later steps in our investigation.

To focus our investigation on the words occurring in the immediate vicinity of *gift**, we sampled 51-word windows of text around the instances of *gift** and *only* respectively, with 25 word tokens before and 25 word tokens after the node words. We also sampled 201-word text windows around the same instances of *gift** and *only*, with 100 word tokens before and 100 word tokens after the node words for use during a subsequent manual review step, reported below. This extraction process yielded 9,151 text windows centered on occurrences of *gift** and 146,516 text windows centered on occurrences of *only* in the COCA fiction subcorpus covering the years 1990–2017. The overall sample constructed for the analyses reported below consists of two sets of $N$ = 155,667 short text windows, with a mean word token count per text window of 50.92 ($SD$ = 1.12) and 199.22 ($SD$ = 10.50) word tokens for the 51-word and 201-word samples respectively. The slightly lower word counts for the average lengths of the passages reflect the fact that some texts contained fewer than the targeted 51 or 201 word tokens when the node words (i.e., *gift** or *only*) were positioned towards the beginning or end of a given text.

Third, we then limited the *gift** corpus to those text windows reflecting the two usages of the lexeme "gift" related to the concept of giftedness in the online edition of *Merriam-Webster's Dictionary* [78]: sense 1 (of three) listed for the noun lemma *gift* ("a notable capacity, talent, or endowment") and sense 1 (of two) listed for the verb lemma *gift* ("to endow with some power, quality, or attribute"). The usages unrelated to giftedness (e.g., text windows referring to birthday gifts) were removed in a manual coding process in which the first author read the 51-word text windows centered on *gift** to determine whether each passage was indicative of giftedness according to the aforementioned definitions. When the 51-word text windows provided insufficient context for determining the sense of the lexeme "gift," the longer 201-word text windows were consulted to allow for the consideration of additional context. We assessed the reliability of our manual coding process via a comparison with a manual coding process implemented by an external rater working independently. Interrater agreement [79] between the internal and external coding was high. Interrater agreement defined as the percentage of agreements was 93%, indicating consistency between our coding and the external coding. Chance-corrected interrater agreement, Cohen's κ, was .82 (95% CI [.73, .91]). See the Technical Supplement (S1 File) for additional details on the manual coding procedure and the test of interrater agreement. The final giftedness-related *gift** corpus then consisted of 2,104 giftedness-related text windows centered on *gift**.

Fourth, we then pre-processed the resulting *gift** and *only* corpora, following standard data-cleaning and data-preparation practices in quantitative text analysis investigations of semantic meaning [80, 81]. We systematically removed left-over HTML markup, resolved contractions to full forms (e.g., "I'd" was resolved to "I had"), and standardized irregular or compound instances of 'gift' and 'only' (e.g., "giftthe" was resolved to "gift the"). Due to the subsequent removal of common grammar words, the resolution of ambiguous contractions to one possible full form was a sufficient solution, because the resulting words were then

removed. For example, whether "I'd" was resolved to "I had" or "I would" was inconsequential as "had" and "would" were both subsequently removed.

In order to focus our investigation on words with semantic meaning, we then used an existing stop-word list [82] (see therein online appendix 11) to remove punctuation, numbers, non-alphanumeric characters, uniform resource locators, and 571 common grammar words. During this step, we also removed the node words *gift\** and *only* from the respective samples. Finally, we lemmatized the word tokens in the text windows to consolidate grammatical inflections into their base forms, known as lemmata. For example, the word types *say* and *said* were combined and each such word token was uniformly represented by the base-form word type *say*.

The final *gift\** corpus consisted of 2,104 text windows centered on *gift\** with a mean length of 14.52 words ($SD$ = 4.17); the final *only* corpus consisted of 146,516 text windows centered on *only* with a mean length of 15.23 words ($SD$ = 4.19). Additional adjustments to the corpora as required by the analytical procedures used to analyze the corpora are described below. Table 1 illustrates how the preprocessing steps just described as well as the adjustments made when working with the samples (reported below) altered the samples.

For a more detailed description of the process of constructing and analyzing our US fiction subcorpus, refer to the Technical Supplement (S1 File) and our repository with data files produced during all steps of the analysis below (https://epub.uni-regensburg.de/58477/). The Technical Supplement includes a detailed step-by-step description of how we derived our samples from the COCA and prepared these for each analytical step in our study. It reproduces our entire code pipeline and explains and documents all of the functions and tools we used for the entire study. Due to licensing restrictions, we were unable to include the original COCA data in the repository. However, we explain in the supplement how users can license and download the original data.

## Analytical procedures and measurement instrument

In the following, we will describe the analytical procedures we implemented to investigate RQs 1 and 2. For investigating RQ 1, we performed a keyword identification procedure and a principal component analysis. The keyword identification procedure provides a general sense of the salient word types in the vicinity of the node word *gift\**; the principal component analysis provides insights into salient clusters of words (i.e., themes) occurring near *gift\**. For investigating RQ 2, we performed a sentiment analysis, which allows us to assess the emotional tone of the *gift\** and *only* corpora. The three procedures required different additional preprocessing steps for the respective corpora. Below we describe the ways in which the corpora were

**Table 1. Illustration of how preprocessing affects the composition of a text window in the *gift\** corpus.**

| Step | Word tokens preceding node word | Node | Word tokens following node word |
|---|---|---|---|
| 1 | Carla was a gifted person. I do n't mean in the usual ways of voice or hand or brain or talent. She was | gifted | in the faculty for love. It came from her naturally, as fragrance comes from a flower. Once, early in our love |
| 2 | Carla was a gifted person. I do n't mean in the usual ways of voice or hand or brain or talent. She was | gifted | in the faculty for love. It came from her naturally, as fragrance comes from a flower. Once, early in our love |
| 3 | Carla was a gifted person. I do n't mean in the usual ways of voice or hand or brain or talent. She was | gifted | in the faculty for love. It came from her naturally, as fragrance comes from a flower. Once, early in our love |
| 4 | Carla was a gifted person. I do n't mean in the usual ways of voice or hand or brain or talent. She was | gifted | in the faculty for love. It came from her naturally, as fragrance comes from a flower. Once, early in our love |

subsequently modified in conjunction with each analytical procedure. For replication of the three analytical procedures, readers are also directed to the Technical Supplement (S1 File), which describes in detail, down to the level of the implemented code, how we implemented the analytical procedures.

Our first step in investigating RQ 1 involved keyword identification. The keyword identification procedure uses a comparison between a target corpus (i.e., the body of texts we intend to investigate) and a reference corpus (i.e., a corpus used as a baseline or comparison) to identify salient words in the target corpus in comparison to the reference corpus. When a word type occurs in the target corpus at a rate higher than what would be expected by chance based on the relative frequency of that word type in the reference corpus, one speaks of positive keywords in the target corpus [83, 84]. For our investigation, the *gift\** corpus described above was designated as the target corpus, and the *only* corpus was designated as the reference corpus. As *only* is a common grammar word, it should appear at similar rates throughout the COCA fiction subcorpus [73], thus providing a proxy of language use in general throughout the COCA fiction subcorpus.

The goal for our keyword identification was to identify semantically meaningful words that occurred at unexpectedly high rates in the text windows centered on *gift\** (related to giftedness). Following the established practice of corpus linguistics investigations of word meaning by measures of association [77, 80, 84], such findings suggest common associations related to the node word in the target corpus, in our case related to giftedness. Thus, findings from the keyword identification process should provide an initial answer related to RQ 1.

To avoid spurious findings, we limited our keyword investigation to word tokens that occurred in both the target and the reference corpora at least 10 times. This additional preprocessing step reflects standard practice in quantitative text-analysis investigations of semantic meaning [80, 81]. For implementing the keyword identification procedure, we then fashioned a matrix with one row for the target corpus (i.e., the *gift\** passages), one row for the reference corpus (i.e., the *only* passages), and 84,037 columns (see the Technical Supplement). The cell values for columns 1–84,037 record the number of word tokens (i.e., counts or occurrences) for all the word types occurring in one or both of the corpora at least 10 times (i.e., columns correspond to word types). The matrix values provide a contingency table that we then assess using an inferential statistical measure to identify word types in the target corpus that occur at rates that are, statistically speaking, higher than what would be expected based on their occurrences in the reference corpus. Words thus identified in the target corpus are considered *positive keywords* and provide insight into the semantic environment of the writing in the immediate vicinity of *gift\** in the sampled fictional texts.

**Principal component analysis.** Still with respect to RQ 1, we then implemented a principal component analysis (PCA) to identify salient clusters of words (i.e., themes) occurring in the vicinity of the node word *gift\**. PCA is a dimension-reduction technique used to identify patterns or themes in data sets by identifying components that explain the greatest amount of variance in the data [85]. PCA is used in a variety of fields. In quantitative text analysis, it is an established technique for revealing patterns of meaning in documents once the documents have been transformed into a document–feature matrix [81, 86]. Due to the nature of PCA, which does not rely on a comparison with another data set, the reference corpus (i.e., the *only* passages) was not needed. While various non-deterministic topic-modeling procedures such as latent Dirichlet allocation are frequently employed for identifying salient clusters of words within unstructured language data [87], we chose PCA for our exploratory investigation of potential themes occurring in the vicinity of *gift\** (in the sense of giftedness) in our fiction subcorpus to support the replicability of our procedure. The PCA method is deterministic [81], which ensures that repeated implementations of the method on a given data set will yield

identical findings. Moreover, familiarity with factor-analytical procedures is widespread in general [88], and the employment of PCA as a means of identifying themes within unstructured text data is an established and well-documented practice in psychology and allied fields [81, 86].

Following recommendations for use of PCA in quantitative text analysis [81, 86], we subjected our *gift** corpus to three additional preparatory steps before running the PCA. First, we converted it to a document–feature matrix, in which each row corresponds to one of the 2,104 text windows centered on *gift** described above, each column represents one of the word types (i.e., features) occurring in at least one of the texts, and each cell records the number of occurrences of a given word type (column) in a given document (row). The resulting matrix includes 8,707 columns, each corresponding to one word type.

Second, we limited the columns to word types that occur in at least 34 different documents (i.e., matrix rows), which means that we introduced an inclusion threshold for word types, requiring that a word type occur in at least 1.616% of the 2,104 *gift** passages. We arrived at this inclusion threshold following recent guidance on using PCA to identify themes in texts [81]. Determining the inclusion threshold must strike a balance between excluding words that appear in low percentages of texts (and thereby contribute little to meaningful themes) and maintaining a great enough variety of word types in the document–feature matrix to ensure that the subsequent analysis will still yield a "meaningful and interpretable number of themes" ([81], p. 5). See Section 10.1 in the Technical Supplement (S1 File) for details on how we determined the inclusion threshold.

This preprocessing step substantially reduced the matrix, leaving only 93 word types (i.e., columns). The removal of low-frequency word types (defined as word types that occurred in fewer than 34 of the 2,104 documents included in the *gift** corpus) lowered the length of the processed text windows centered on *gift** (as reported above) to a mean length of 3.12 words ($SD$ = 1.826). This means that by excluding rare words, the filtered document–feature matrix focuses the PCA on frequent word types. The frequent word types retained in each document originally occurred over a range—considering the original sampling window, prior to all preprocessing steps, with all the excluded words—of 25 words before and 25 words after the node word *gift**. For an illustration of how the remaining (frequent) word types are distributed across the original text data, see Table 1 (specifically Step 4 therein). Overall, the filtered document–feature matrix consists of 2,104 documents represented by the rows and 93 word types (i.e., features) represented by the columns. The matrix's 195,672 cells (see Table 2) record word counts per document. This indicates how often a given word type (column) occurs in a given document (row). In 189,483 cells, the word count is 0, indicating that the word type does not occur in that document. In 5,843 cells, the word count is 1, indicating that the word type occurs once in that document. In 329 cells, the word count is 2, indicating that the word type occurs twice in that document. In 12 cells, the word count is 3, indicating that the word type occurs three times in that document. Finally, in five cells, the word count is 4, indicating that the word type occurs four times in that document.

**Table 2. Word count frequencies of the PCA matrix.**

| Word count | Number of cells |
| --- | --- |
| 0 | 189,483 |
| 1 | 5,843 |
| 2 | 329 |
| 3 | 12 |
| 4 | 5 |

The word counts are derived from the filtered text passage–word type pre-binarization document–feature matrix, which has 2,104 rows, 93 columns, and 195,672 cells. The matrix was created for the principal component analysis (PCA). The word counts in the matrix indicate how often a given word type (matrix column) occurs in a given document (matrix row).

Third, we binarized the matrix by converting all non-zero values to 1. Binarization is recommended as a standard practice [81] when preparing unstructured text data for examination via PCA, as word types in text corpora based on natural language use often have a modal value of 0 for occurrences per word type [89].

Finally, after preparing the text data for PCA but before running the PCA, we implemented a parallel analysis [90, 91] to ascertain the optimal number of factors for inclusion in the PCA. Parallel analysis is considered the "gold standard technique" [92] (p.8) for specifying a number of factors or components to be included in factor-analytical techniques.

**Sentiment analysis.** For our sentiment analysis, we used existing lookup dictionaries for positive and negative emotions included in the Lexicoder Sentiment Dictionary (LSD) [71] to interrogate our *gift** and *only* corpora regarding the affective tone of the texts with respect to RQ 2. The LSD was developed primarily as a sentiment analysis tool for political communication. However, we deemed it a good choice for our investigation based on its widespread implementation and availability within quantitative text analysis [93], its methodologically thorough incorporation of earlier sentiment-analysis dictionaries that are available without proprietary restrictions and thus readily auditable in future research, the developers' assertion that the LSD was also designed for implementation in other domains, and, finally, its extensive external validation against other widespread sentiment-analysis dictionaries as well as against human coding [71]. The lookup dictionaries reflecting positive and negative affect in the LSD incorporated the lists of positive and negative words found in a prominent thesaurus [94], an early lookup dictionary designed for quantitative text analysis [95], and a lookup dictionary designed for the quantitative psychological investigation of literature that included registers associated with negative and positive emotions [96, 97].

For the sentiment analysis, we used a non-lemmatized version of the pre-processed corpus, thus retaining word types' grammatical inflections (e.g., *kitten* and *kittens*), because the LSD includes wildcards allowing words in the dictionary to match with various forms. For example, the dictionary listing *virtue** will match *virtue* and *virtues*, and the dictionary listing *virtuo** will match *virtuous*, *virtuousness*, *virtuosity*, and *virtuoso*. This modification of the corpora did not affect the mean length and standard deviations of the *gift** and *only* corpora; they remain as reported in the sample description above.

## Results

### Research Question 1 (RQ1)

RQ 1 asked which notions of giftedness and the gifted are salient in literary texts published in recent decades in the United States. We report the findings of methodologically complementary investigations of RQ 1. First, the keyword identification procedure provided insights into salient words in the gifted passages. Second, the principal component analysis (PCA) provided insights into themes within the gifted passages.

**Keyword identification.** The keyword identification procedure identified 92 positive keywords in the text passages centered on *gift** that constitute the target corpus (see Table 3). The keywords are word types that occurred at rates in the target corpus that, when compared with their occurrence rates in the reference corpus, are unlikely to have occurred so frequently by chance. We assessed word keyness by using the log-likelihood ratio $G^2$ with the Williams correction, which is commonly used for assessing the keyness of word types in a target corpus in

**Table 3. Significant lemmatized keywords in the target corpus (*gift**) in comparison to the reference corpus (*only*).**

| Keyword | $G^2$ | $p$ (FDR) | Occurrences TC | Occurrences RC | Normalized occurrences TC | Normalized occurrences RC | Possible topics |
|---|---|---|---|---|---|---|---|
| talent | 485.122 | < .001 | 108 | 270 | 353 | 12 | Ability |
| ability | 169.314 | < .001 | 52 | 265 | 170 | 12 | Ability |
| special | 166.069 | < .001 | 75 | 735 | 245 | 33 | Rarity or prowess |
| great | 154.385 | < .001 | 131 | 2,504 | 429 | 112 | Rarity or prowess |
| power | 120.091 | < .001 | 82 | 1,290 | 268 | 58 | Ability |
| possess | 105.757 | < .001 | 40 | 292 | 131 | 13 | Ability |
| god | 92.926 | < .001 | 106 | 2,523 | 347 | 113 | Etiology |
| magic | 84.954 | < .001 | 41 | 438 | 134 | 20 | The metaphysical realm |
| skill | 83.608 | < .001 | 35 | 297 | 115 | 13 | Ability |
| language | 81.830 | < .001 | 49 | 674 | 160 | 30 | Arts and letters |
| writer | 73.955 | < .001 | 33 | 306 | 108 | 14 | Arts and letters |
| art | 71.498 | < .001 | 44 | 623 | 144 | 28 | Arts and letters |
| natural | 70.442 | < .001 | 42 | 575 | 137 | 26 | Etiology |
| psychic | 69.258 | < .001 | 18 | 60 | 59 | 3 | The metaphysical realm |
| healer | 65.012 | < .001 | 14 | 27 | 46 | 1 | Medicine |
| extraordinary | 57.656 | < .001 | 19 | 105 | 62 | 5 | Rarity or prowess |
| artist | 53.739 | < .001 | 26 | 266 | 85 | 12 | Arts and letters |
| music | 50.904 | < .001 | 44 | 853 | 144 | 38 | Arts and letters |
| genius | 47.094 | < .001 | 18 | 128 | 59 | 6 | Rarity or prowess |
| spirit | 45.531 | < .001 | 33 | 549 | 108 | 25 | The metaphysical realm |
| curse | 48.065 | < .001 | 25 | 302 | 82 | 14 | Burden of giftedness |
| rare | 44.949 | < .001 | 26 | 330 | 85 | 15 | Rarity or prowess |
| bestow | 44.690 | < .001 | 12 | 41 | 39 | 2 | Etiology |
| prophecy | 43.787 | < .001 | 12 | 43 | 39 | 2 | The metaphysical realm |
| musical | 43.758 | < .001 | 14 | 71 | 46 | 3 | Arts and letters |
| student | 41.983 | < .001 | 41 | 874 | 134 | 39 | Academics and learning |
| poet | 40.522 | < .001 | 20 | 207 | 65 | 9 | Arts and letters |
| athlete | 39.208 | < .001 | 11 | 41 | 36 | 2 | Sports |
| powerful | 37.905 | < .001 | 23 | 306 | 75 | 14 | Rarity or prowess |
| learn | 36.698 | < .001 | 59 | 1,722 | 193 | 77 | Academics and learning |
| speech | 36.648 | < .001 | 21 | 261 | 69 | 12 | Arts and letters |
| exceptional | 34.267 | < .001 | 10 | 40 | 33 | 2 | Rarity or prowess |
| bless | 33.859 | < .001 | 16 | 154 | 52 | 7 | Etiology |
| develop | 33.639 | < .001 | 22 | 316 | 72 | 14 | Etiology |
| beauty | 33.554 | < .001 | 27 | 493 | 88 | 22 | Beauty |
| master | 32.979 | < .001 | 31 | 642 | 101 | 29 | Rarity or prowess |
| nurture | 32.761 | < .001 | 10 | 44 | 33 | 2 | Etiology |
| blessing | 32.314 | < .001 | 14 | 118 | 46 | 5 | Etiology |
| wonderful | 32.299 | < .001 | 20 | 270 | 65 | 12 | Rarity or prowess |
| marvelous | 32.057 | < .001 | 10 | 46 | 33 | 2 | Rarity or prowess |
| mage | 31.333 | < .001 | 13 | 102 | 43 | 5 | The metaphysical realm |
| grateful | 31.163 | < .001 | 17 | 197 | 56 | 9 | Unclear |
| goddess | 30.484 | < .001 | 12 | 86 | 39 | 4 | The metaphysical realm |
| fairy | 30.339 | < .001 | 14 | 129 | 46 | 6 | The metaphysical realm |
| naturally | 29.682 | < .001 | 16 | 182 | 52 | 8 | Etiology |
| amazing | 29.352 | < .001 | 15 | 159 | 49 | 7 | Rarity or prowess |
| grant | 27.950 | < .001 | 21 | 361 | 69 | 16 | Etiology |
| school | 27.884 | < .001 | 70 | 2,509 | 229 | 112 | Academics and learning |
| sensitive | 27.802 | < .001 | 12 | 99 | 39 | 4 | Psychosocial ability |

(*Continued*)

**Table 3.** (Continued)

| Keyword | $G^2$ | $p$ (FDR) | Occurrences TC | Occurrences RC | Normalized occurrences TC | Normalized occurrences RC | Possible topics |
|---|---|---|---|---|---|---|---|
| highly | 27.774 | < .001 | 14 | 145 | 46 | 6 | Rarity or prowess |
| sight | 27.209 | < .001 | 33 | 817 | 108 | 37 | Psychosocial ability |
| nature | 26.984 | < .001 | 24 | 476 | 79 | 21 | Etiology |
| surgeon | 26.648 | < .001 | 11 | 84 | 36 | 4 | Medicine |
| divine | 26.039 | < .001 | 11 | 87 | 36 | 4 | Etiology |
| incredible | 24.700 | < .001 | 11 | 94 | 36 | 4 | Rarity or prowess |
| prove | 24.177 | < .001 | 26 | 594 | 85 | 27 | Unclear |
| earn | 23.773 | .001 | 17 | 263 | 56 | 12 | Etiology |
| teacher | 23.491 | .001 | 25 | 567 | 82 | 25 | Academics and learning |
| make | 23.319 | .001 | 257 | 13,646 | 841 | 612 | Unclear |
| vision | 23.083 | .001 | 21 | 424 | 69 | 19 | Psychosocial ability |
| beautiful | 23.062 | .001 | 35 | 990 | 115 | 44 | Beauty |
| mother | 22.638 | .002 | 130 | 6,043 | 425 | 271 | Etiology |
| smart | 22.605 | .002 | 20 | 395 | 65 | 18 | Ability |
| way | 22.451 | .002 | 25 | 585 | 82 | 26 | Unclear |
| class | 22.053 | .002 | 36 | 1,059 | 118 | 47 | Academics and learning |
| people | 21.686 | .002 | 144 | 6,925 | 471 | 310 | Unclear |
| believe | 21.090 | .003 | 32 | 905 | 105 | 41 | Unclear |
| mortal | 20.621 | .004 | 10 | 95 | 33 | 4 | Unclear |
| young | 20.141 | .005 | 77 | 3,200 | 252 | 143 | Youth |
| potential | 19.739 | .006 | 12 | 153 | 39 | 7 | Ability |
| charm | 19.163 | .008 | 12 | 158 | 39 | 7 | Psychosocial ability |
| program | 19.132 | .008 | 18 | 373 | 59 | 17 | Academics and learning |
| understanding | 18.941 | .008 | 10 | 106 | 33 | 5 | Academics and learning |
| inherit | 18.829 | .008 | 12 | 161 | 39 | 7 | Etiology |
| pride | 18.600 | .009 | 16 | 291 | 52 | 13 | Emotions |
| write | 18.517 | .010 | 54 | 2,049 | 177 | 92 | Arts and letters |
| waste | 18.473 | .010 | 18 | 383 | 59 | 17 | Ability forsaken |
| addition | 17.971 | .012 | 10 | 113 | 33 | 5 | Unclear |
| life | 17.700 | .014 | 122 | 5,915 | 399 | 265 | Unclear |
| bear | 17.408 | .016 | 43 | 1,531 | 141 | 69 | Burden of giftedness |
| world | 17.310 | .016 | 91 | 4,137 | 298 | 185 | The physical realm |
| study | 16.981 | .019 | 31 | 965 | 101 | 43 | Academics and learning |
| give | 16.376 | .025 | 96 | 4,484 | 314 | 201 | Unclear |
| heal | 16.303 | .025 | 12 | 186 | 39 | 8 | Medicine |
| tongue | 16.085 | .028 | 20 | 503 | 65 | 23 | Arts and letters |
| training | 15.997 | .029 | 13 | 222 | 43 | 10 | Etiology |
| human | 15.979 | .029 | 49 | 1,893 | 160 | 85 | The physical realm |
| choose | 15.791 | .031 | 30 | 952 | 98 | 43 | Unclear |
| future | 15.489 | .036 | 25 | 731 | 82 | 33 | Unclear |
| influence | 15.435 | .037 | 10 | 134 | 33 | 6 | Unclear |
| age | 15.269 | .040 | 38 | 1,357 | 124 | 61 | Unclear |
| intelligence | 15.152 | .041 | 12 | 199 | 39 | 9 | Ability |

TC = target corpus (i.e., the text passages around the node word *gift**); RC = reference corpus (i.e., the text passages around the node word *only*); FDR = false-discovery rate. The target and reference corpus columns list the absolute occurrences of the lemmatized word in the target and reference corpus respectively. The normalized target and reference corpus columns reflect normalized values of the same occurrences at 100,000 word tokens. The possible topics column lists possible themes—formulated *prima facie* by the authors—suggested by the keywords of *gift**. As such, the proposed themes are exploratory. For each proposed theme, subsequent investigations would be needed to confirm the appropriateness of the initial thematic interpretations.

comparison to a reference corpus [84, 98, 99]. The log-likelihood ratio $G^2$ does not require the assumption of normality to be fulfilled; and the Williams correction reduces type-I errors (i.e., false positives) that may occur due to small sample sizes or for low-frequency word types. We additionally corrected the resulting $p$ values with the false-discovery-rate (FDR) correction [100] to account for the distorting effect of making multiple comparisons.

The keywords cover a variety of topics and are semantically distinct in most cases. The 21 keywords with the greatest $G^2$ values, for example, recall notions of ability (*talent*, *ability*, *power*, *possess*, *skill*, and *genius*), rarity or prowess (*special*, *great*, *healer*, and *extraordinary*), the etiology of giftedness (*god* and *natural*), the metaphysical realm (*god*, *magic*, *psychic*, *healer*, *spirit*, and *curse*), and arts and humanities domains (*language*, *writer*, *art*, *artist*, and *music*). The FDR-corrected $p$ values for the first 21 keywords suggest that these keywords are strongly associated with the *gift** target corpus; the FDR-corrected $p$ value of the twentieth keyword, *genius* ($p$ = 0.00000002843), for example, lies 5.551 sigma (σ) beyond the middle of the FDR-corrected $p$-value distribution. The level of certainty that our result is not merely reflecting chance can be viewed as very high. It exceeds, for example, the 5-σ gold standard used in particle physics experiments [101]. We estimated the noted semantic distinctness of the keywords via the extent of semantic overlap among the keywords by fetching each keyword's synonyms in the Princeton Wordnet lexical database [102], counting the number of shared synonyms among all pairs of keywords, and recording the pair-wise shared synonym counts in a similarity matrix (see Fig 2 and the Technical Supplement). Among the 92 keywords, 39 share synonyms with one or more other keyword. However, the synonym overlap among the keywords is small overall. Of the 8,464 pairs of keywords, only 102 share one or more synonyms, and among these, the modal value of shared synonyms is 1.

The 92 positive keywords reported in Table 3 provide a sense of the concerns the authors of the US fictional texts were addressing when they used the term *gift** in the sense of giftedness. Taken together, the positive keywords suggest that the authors of the fictional texts associated giftedness with a range of concerns such as the following: ability (*talent*, *ability*, *power*, *possess*, *skill*, *smart*, *potential*, and *intelligence*), ability forsaken (*waste*), academics and learning (*student*, *learn*, *school*, *teacher*, *class*, *program*, *understanding*, and *study*), arts and letters (*language*, *writer*, *art*, *artist*, *music*, *musical*, *poet*, *speech*, *write*, and *tongue*), beauty (*beauty* and *beautiful*), the burden of giftedness (*curse* and *bear*), emotions (*pride*), the etiology of giftedness (*god*, *natural*, *bestow*, *bless*, *develop*, *nurture*, *blessing*, *naturally*, *grant*, *nature*, *divine*, *earn*, *mother*, *inherit*, and *training*), medicine (*healer*, *surgeon*, and *heal*), psychosocial ability (*sensitive*, *sight*, *vision*, and *charm*), rarity or prowess (*special*, *great*, *extraordinary*, *genius*, *rare*, *powerful*, *exceptional*, *master*, *wonderful*, *marvelous*, *amazing*, *highly*, and *incredible*), sports (*athlete*), the metaphysical realm (*magic*, *psychic*, *spirit*, *prophecy*, *mage*, *goddess*, and *fairy*), the physical realm (*world* and *human*), and youth (*young*). While aspects of education-stakeholder-focused giftedness inquiry [19, 23] such as ability, academics and learning, and psychosocial ability are present, the keywords derived from fictional texts reflect a wider ambitus of associated notions. Our prima facie interpretation and categorization of the positive keywords associated with the use of the word *gift**—all instances of which, as noted above, were within the dictionary senses related to the concept of giftedness—provides an initial answer related to RQ 1 regarding the notions of giftedness and the gifted common in US fictional texts.

**Principal components analysis (PCA).** The second step in investigating RQ 1 was the identification of themes via a principal component analysis (PCA). The binarized matrix described above was deemed suitable to investigation via PCA. To assess the appropriateness of PCA for the data set, we calculated the Kaiser-Meyer-Olkin measure of sampling adequacy and Bartlett's test of sphericity. At 0.501, the Kaiser-Meyer-Olkin statistic is below the threshold of 0.6 commonly advised for factor analysis [85]. However, a lower threshold can be

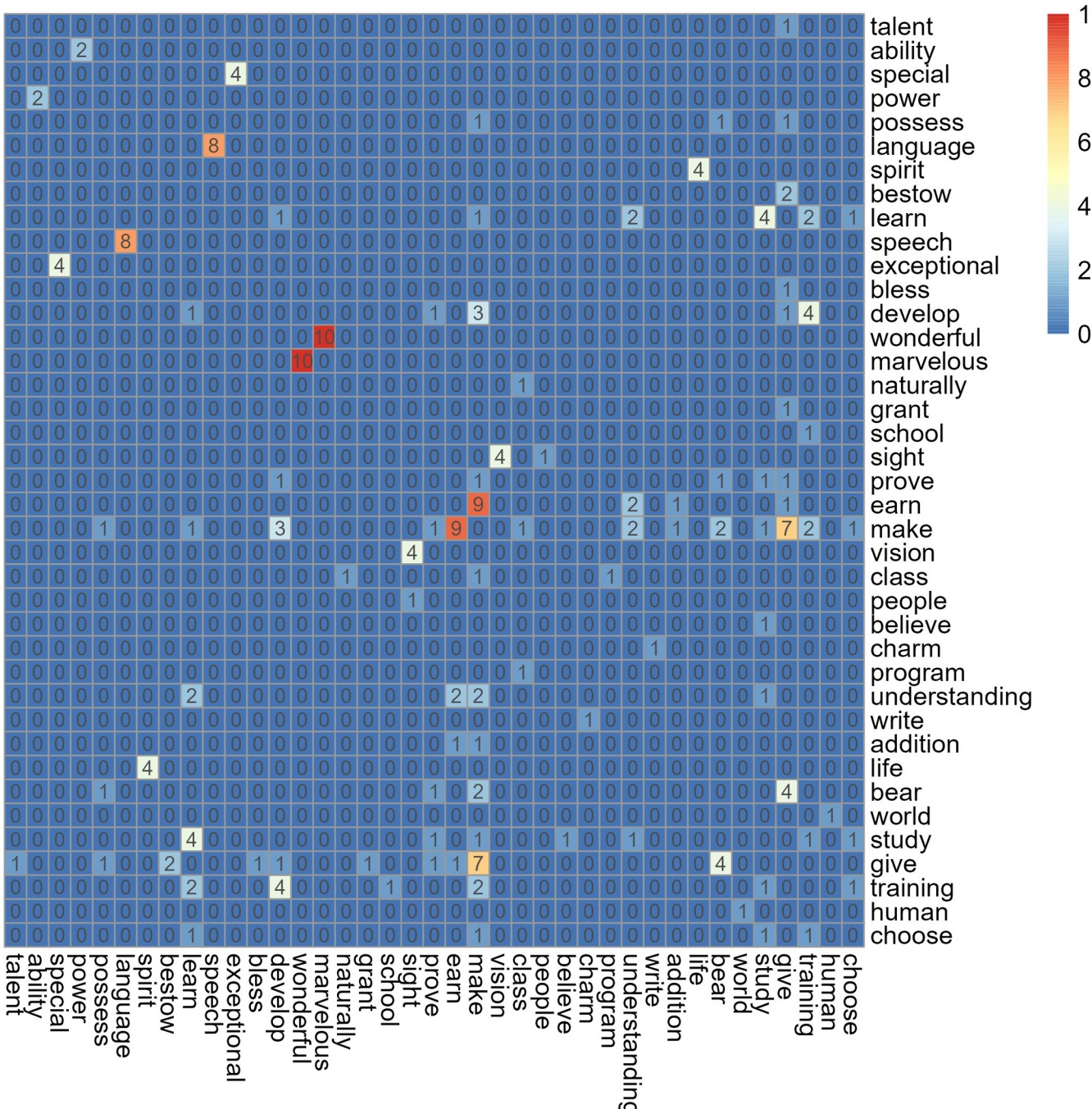

**Fig 2. Heatmap showing all keywords in the similarity matrix with overlapping synonyms among their respective synonyms in the Princeton Wordnet lexical database.** The similarity matrix shows the extent of semantic overlap among the 39 keywords that share one or more synonyms. The numbers in the cells represent the number of shared synonyms between these keywords.

permissible when using PCA to investigate language data [81]. Bartlett's test of sphericity was calculated to confirm that the matrix was not an identity matrix, which would indicate a lack of intercorrelations among the matrix components and thus its unsuitability for PCA. The test results ($\chi^2(4,278) = 5,623.301$, $p < 0.001$) suggest that the binary matrix is amenable to examination via PCA.

Before running the PCA, we implemented a parallel analysis with the default minimum factoring method [103] to ascertain the optimal number of components for extraction in our PCA. As the parallel analysis is a non-deterministic function, we ran it 100 times and then calculated the modal value (mode = 20) for the 100 suggested numbers of components for our PCA. To extract statistically independent themes, we implemented the PCA using varimax rotation, as orthogonal rotations are advised for implementations of factor analysis in quantitative text analysis [81, 86].

Overall, the 20-component solution explained 29.083% of the variance among all variables (i.e., word types) in the binarized document–feature matrix. For each component, we report in Table 4 the loadings greater than or equal to .2, which reflects a common cutoff for investigations of themes in quantitative text data via PCA [81]. Considering the component-specific loadings as of this threshold, 11 components yielded nine readily interpretable themes: formal education (Component 1), youth and boys (Component 14), written communication (Component 10), parents and art (Component 19), potent natural ability (Components 15 and 3), supernatural ability (Component 5), joy (Component 4), verbal communication (Components 20 and 2), and music (Component 7). The interpretable components provide additional insight into RQ 1 that complements the findings from the keyword analysis. In the previous step, the individual keywords reflected corrected log-likelihood ratios and thus have an empirical basis, while the proposed categorization of the keywords into possible larger concerns reflected an interpretive step on the part of the authors. While our descriptors of the word tokens loading onto specific components are also interpretive in nature, the linking of the word types associated with a given component (variables in the PCA) has an empirical basis in the PCA and is replicable.

**Validity check.** For inductive (i.e., bottom-up) methods of quantitative text analysis, which include keyword identification and PCA techniques, it is recommended to consider evidence for the validity of findings [81]. We therefore considered evidence of content validity and discriminant validity [104] for the keyword identification and the PCA.

*Content validity* reflects the extent to which findings are conceptually related to the concept under investigation. The findings reported for the keyword identification procedure and the PCA provide evidence of content validity. The keywords reported in Table 3 are readily interpretable within the context of the noted definition of giftedness used in constructing our sample [78]. The PCA themes reported in Table 4 also stand up to such a review, but not in all cases: Most of the words reported for each component (i.e., those with loadings of .2 or greater) appear plausible within the remit of our chosen dictionary definition of giftedness. However, in the case of the PCA-derived themes, for 9 of the 20 components and their respective words, the content validity was not immediately apparent. In such cases, we entered "unclear" in Table 4 for those themes (components).

*Discriminant validity* is a form of validity related to the comparison of evidence arising from conceptually differing variables or measurements, for which one would expect results to differ systematically in accordance with known conceptual differences. The keyword-identification findings and the PCA findings provide evidence of discriminant validity in this sense. For the keyword findings, the nature of the analytical procedure relies on observing robust differences between (a) expected rates of occurrence of word types based on their actual occurrences in a reference corpus (in our case the *only* corpus), which by its nature reflects a broader baseline of language use, and (b) the actual rates of occurrence of the same word types in a target corpus (in our case the *gift** corpus), which is indicative of a more specific case of language use. The keyword-identification findings show evidence of discriminant validity, because, as hypothesized for the test, the two different sets of data returned systematically different word-usage frequencies (see Table 3).

**Table 4. Principal component analysis results for the *gift*\* corpus.**

| Theme Descriptor | Component | Loadings | Eigenvalues (λ) | Proportion of Variance | Cumulative Variance |
|---|---|---|---|---|---|
| Formal education | 1 | school (0.656), high (0.615), talent (0.377), class (0.340), run (0.234), kid (0.215) | 1.510 | 1.623% | 1.623% |
| Youth and boys | 14 | age (0.513), boy (0.457), student (0.431), young (0.396), year (0.247) | 1.409 | 1.515% | 3.139% |
| Written communication | 10 | write (0.517), class (0.286), time (0.271), read (0.263), special (0.218), student (0.210) | 1.397 | 1.503% | 4.641% |
| Parents and art | 19 | mother (0.533), father (0.449), leave (0.351), art (0.256), learn (0.250) | 1.379 | 1.483% | 6.124% |
| Potent natural ability | 15 | bring (0.413), begin (0.382), world (0.360), power (0.346), great (0.251), human (0.235) | 1.375 | 1.479% | 7.603% |
| Unclear | 11 | room (0.515), friend (0.477), people (0.328), face (0.241), sit (0.238), time (0.213), long (0.209) | 1.366 | 1.469% | 9.072% |
| Supernatural ability | 5 | magic (0.565), family (0.462), bear (0.376), think (0.293) | 1.364 | 1.466% | 10.538% |
| Potent natural ability | 3 | ability (0.562), natural (0.470), possess (0.430), power (0.230) | 1.360 | 1.463% | 12.001% |
| Unclear | 9 | work (0.522), put (0.450), live (0.331), talent (0.263), sit (0.240), true (0.230) | 1.358 | 1.460% | 13.461% |
| Unclear | 6 | find (0.305), open (0.210) | 1.352 | 1.454% | 16.369% |
| Unclear | 8 | life (0.450), end (0.428), night (0.297), live (0.291), long (0.274), year (0.263) | 1.352 | 1.454% | 14.915% |
| Unclear | 17 | hand (0.391), woman (0.238), tell (0.219), look (0.202) | 1.340 | 1.441% | 17.810% |
| Joy | 4 | laugh (0.556), smile (0.466), make (0.285), run (0.253), back (0.202) | 1.333 | 1.433% | 19.243% |
| Verbal communication | 20 | talk (0.419), word (0.378), people (0.351), speak (0.268), tell (0.203) | 1.329 | 1.429% | 20.672% |
| Verbal communication | 2 | understand (0.490), story (0.407), true (0.384), tell (0.324), ask (0.237), inside (0.222) | 1.322 | 1.422% | 22.094% |
| Unclear | 13 | place (0.477), show (0.402), child (0.309), read (0.296), woman (0.264), understand (0.239) | 1.319 | 1.418% | 23.513% |
| Unclear | 12 | find (0.294), good (0.281), night (0.279), give (0.212), leave (0.210) | 1.304 | 1.402% | 24.915% |
| Music | 7 | play (0.523), music (0.479), love (0.327), learn (0.236) | 1.301 | 1.399% | 26.314% |
| Unclear | 18 | call (0.312), kind (0.267), ask (0.257), open (0.222) | 1.298 | 1.396% | 27.709% |
| Unclear | 16 | real (0.350), long (0.317), open (0.270), time (0.207) | 1.277 | 1.373% | 29.083% |

The loadings express the strength of the correlation of each word type and a given component, thereby characterizing the extent to which a given word type is contributing to the thematic identity of a component. All loadings greater than or equal to .2 are reported. The eigenvalues (λ), according to which the table is sorted, reflect the amount of variance that a specific principal component explains by accounting for how much each original variable contributes to that component as the sum of the squared component loadings. The proportion of variance explained by a given component is the proportion of the explained variance of all variables (word types) in the PCA. The cumulative variance describes the cumulative variance explained by all components up to and including the current component as a proportion of the variance of all variables (word types) in the PCA. The theme descriptors were ascribed to the components by the authors based on their interpretation of the word tokens with high loadings for a given component.

For the PCA findings, we assessed evidence of discriminant validity in the case of those components for which the words reported in component-specific factor loadings in Table 4 suggested readily interpretable themes (i.e., those reported above and for which evidence of content validity had been found). To this end, we created lookup dictionaries (i.e., word lists) for the nine readily interpretable themes across 11 components as listed in Table 4, using the words reported for each of these components in Table 4, which were based on all factor loadings per component greater than or equal to .2. We then assessed the prevalence of each theme-specific lookup dictionary in our *gift*\* corpus and our *only*\* corpus and then compared the mean prevalence for each theme by corpus type. Here, prevalence reflects the relative frequency of occurrences of the terms comprising a given thematic dictionary by sample in the

*gift** or *only* corpus. We reasoned that if the PCA-derived themes are indeed indicative of giftedness-related themes in the US sample of fictional texts we were investigating, we should see higher relative occurrence rates (i.e., greater prevalence) for the respective dictionaries in the *gift** corpus in comparison to the *only* corpus. Significant differences in prevalence of a given thematic dictionary would provide evidence of discriminant validity.

Before implementing the means comparisons, we assessed the distributional properties of our data. We assessed the normality of the data with the Kolmogorov–Smirnov test. Due to the large sample size, we included the Lilliefors Correction. We then assessed the equality of variances (homoscedasticity) between groups with Levene's test. Consistent with typical distributional patterns of natural language data [89], the test results indicated deviations from normal distributions and from homoscedasticity for all variables, as reported in Table 5.

We then compared the theme-specific prevalence means with Welch's *t* test, a parametric test that does not require equal variances, as well as with the Mann–Whitney U test, a non-parametric test that requires neither equal variances nor normal distributions. The parametric test should be sufficient despite the violation of normality in light of the large sample size. The Mann–Whitney U test adds additional confidence that the test statistic is not merely reflecting artifacts of the data. We assessed the effect sizes of the differences with Cohen's *d* as well as with Cliff's *d*, which does not require a specific shape or spread of the distribution and thereby provides a more conservative supplementary estimate of effect size [105]. As summarized in Table 6, the means comparisons indicated a significantly higher mean prevalence in the *gift** corpus samples in comparison to the *only* corpus samples for eight of nine readily interpretable PCA-derived themes and, therefore, evidence of discriminant validity for these themes. In the case of the 'Joy' theme, the evidence is inconclusive. While the reported effect sizes suggest very slight differences (see Table 6), we deemed the small differences acceptable evidence in the context of exploratory research.

**Synthesis of findings.** Taken together, the findings described for both approaches to investigating the notions of giftedness and the gifted in US fictional texts (RQ 1) can be grouped into four areas: (a) ability, learning and formal education, and the rarity of giftedness, (b) causes and consequences of giftedness, (c) domain associations, and (d) the aesthetic quality of beauty. In Table 7, we categorized our keyword and PCA findings into these four broad

**Table 5. Tests for normality and equality of variances (homoscedasticity) for the cross-validation of PCA-generated themes.**

| Theme | Normality | | | | Equality of Variances | |
|---|---|---|---|---|---|---|
| | *Gift** corpus | | *Only* corpus | | | |
| | *D* | *p* | *D* | *p* | *F* | *p* |
| Formal education | 0.492 | < .001 | 0.520 | < .001 | 142.482 | < .001 |
| Youth and boys | 0.454 | < .001 | 0.480 | < .001 | 33.664 | < .001 |
| Potent natural ability | 0.442 | < .001 | 0.511 | < .001 | 453.987 | < .001 |
| Supernatural ability | 0.504 | < .001 | 0.525 | < .001 | 42.552 | < .001 |
| Parents and art | 0.431 | < .001 | 0.446 | < .001 | 17.901 | < .001 |
| Written communication | 0.469 | < .001 | 0.491 | < .001 | 28.623 | < .001 |
| Verbal communication | 0.420 | < .001 | 0.451 | < .001 | 63.325 | < .001 |
| Joy | 0.461 | < .001 | 0.460 | < .001 | 1.406 | .236 |
| Music | 0.503 | < .001 | 0.525 | < .001 | 99.308 | < .001 |

Normality was assessed with the Kolmogorov–Smirnov Test with the Lilliefors Correction. Equality of variances (homoscedasticity) between groups was assessed with Levene's Test; for all group comparisons therein, degrees of freedom (df) was 1, with an error term 148,618.

**Table 6. Means comparisons for PCA-generated themes in the *gift\** and *only* samples.**

| Theme | Mean | | Welch | | | | Mann–Whitney | | Cohen's *d* | Cliff's *d* |
|---|---|---|---|---|---|---|---|---|---|---|
| | *gift\** | *only* | *t* | DF | *p* | CI | *t* | DF | | |
| Formal education | 0.013 | 0.007 | 7.915 | 2,128.929 | < .001 | 0.004–0.008 | 163,295,537.500 | < .001 | 0.35 | 0.07 |
| Youth and boys | 0.019 | 0.014 | 4.914 | 2,146.123 | < .001 | 0.003–0.006 | 160,918,297.000 | < .001 | 0.18 | 0.05 |
| Written communication | 0.016 | 0.013 | 4.657 | 2,148.638 | < .001 | 0.002–0.006 | 160,546,267.000 | < .001 | 0.40 | 0.08 |
| Parents and art | 0.024 | 0.020 | 3.806 | 2,151.848 | < .001 | 0.002–0.006 | 159,127,557.500 | < .001 | 0.14 | 0.03 |
| Potent natural ability | 0.022 | 0.009 | 12.697 | 2,123.735 | < .001 | 0.011–0.015 | 173,635,629.000 | < .001 | 0.09 | 0.03 |
| Supernatural ability | 0.010 | 0.006 | 4.643 | 2,133.058 | < .001 | 0.002–0.005 | 158,678,049.000 | < .001 | 0.11 | 0.04 |
| Joy | 0.018 | 0.017 | 1.033 | 2,148.766 | .301 | -0.001–0.003 | 152,618,100.500 | 0.07 | 0.06 | 0.02 |
| Verbal communication | 0.027 | 0.020 | 6.709 | 2,145.703 | < .001 | 0.005–0.010 | 164,055,625.000 | < .001 | 0.14 | 0.04 |
| Music | 0.011 | 0.006 | 7.521 | 2,136.947 | < .001 | 0.004–0.006 | 163,680,797.000 | 0.03 | 0.05 | 0.02 |

categories to provide a more succinct overview of findings pertaining to RQ 1. This reflects an interpretation of the individual findings reported in Tables 3 and 4.

## Research Question 2 (RQ2)

RQ 2 asked about the overall sentiment valence when the terms *giftedness/gifted* are used in fictional texts. It was investigated via sentiment analysis. The interrogation of our *gift\** and *only* corpora using lookup dictionaries for positive and negative emotions from the Lexicoder Sentiment Dictionary (LSD) [71] produced prevalence values for positive and negative emotion words in each corpus.

Before implementing means comparisons by corpus, we assessed the distributional properties of our data. We considered the normality of the data with the Kolmogorov–Smirnov test. Due to the large sample size, we included the Lilliefors Correction. We then assessed the equality of variances (homoscedasticity) between groups with Levene's test. Consistent with typical distributional patterns of natural language data [89], the test results indicated deviations from normal distributions for both variables and from homoscedasticity for the prevalence of positive sentiment, but not for the prevalence of negative sentiment, as reported in Table 8.

We then compared the corpus-specific mean prevalence values for positive and negative emotion words with Welch's *t* test, a parametric test that does not require equal variances, as well as with the Mann–Whitney U test, a non-parametric test that requires neither equal variances nor normal distributions. The parametric test should be sufficient despite the violation of normality in light of the large sample size. The Mann–Whitney U test adds additional confidence that the test statistic is not merely reflecting artifacts of the data. We assessed the effect sizes of the differences with Cohen's *d* as well as with Cliff's *d*, which does not require a specific shape or spread of the distribution and thereby provides a more conservative supplementary estimate of effect size [105].

The means comparisons indicated significant differences in prevalence by corpus, with the *gift\** corpus containing a significantly higher mean prevalence for positive emotion words ($m = 0.126$ versus $m = 0.077$) and a significantly lower mean prevalence for negative emotion words ($m = 0.092$ versus $m = 0.099$) in comparison to the *only* corpus, as reported in Table 9. While the reported effect size for the difference in the prevalence of negative emotion words is small (Cohen's $d = -0.069$, Cliff's $d = -0.039$) and thus equivocal, the significant difference should be noted as an initial exploratory finding. The reported effect size for the difference in the prevalence of positive emotion words is substantial (Cohen's $d = 0.595$, Cliff's $d = 0.298$).

**Table 7. Summary of themes.**

| Area | Descriptor | Subsumed Topic | Origin | Terms |
|---|---|---|---|---|
| 1 | Ability, learning and education, and the rarity of giftedness | Ability | Keywords | talent, ability, power, possess, skill, smart, potential, and intelligence |
| | | Academics and learning | Keywords | student, learn, school, teacher, class, program, understanding, and study |
| | | Formal education | PCA, Component 1 | school, high, talent, class, run, and kid |
| | | Potent natural ability | PCA, Component 10 | bring, begin, world, power, great, human, ability, natural, and possess |
| | | Supernatural ability | PCA, Component 5 | magic, family, bear, and think |
| | | Psychosocial ability | Keywords | sensitive, sight, vision, and charm |
| 2 | Causes and consequences of giftedness | The burden of giftedness | Keywords | curse and bear |
| | | Ability forsaken | Keywords | waste |
| | | Emotions | Keywords | pride |
| | | Rarity or prowess | Keywords | special, great, extraordinary, genius, rare, powerful, exceptional, master, wonderful, marvelous, amazing, highly, and incredible |
| | | The etiology of giftedness | Keywords | god, natural, bestow, bless, develop, nurture, blessing, naturally, grant, nature, divine, earn, mother, inherit, and training |
| | | The metaphysical realm | Keywords | magic, psychic, spirit, prophecy, mage, goddess, and fairy |
| | | The physical realm | Keywords | world and human |
| | | Youth | Keywords | youth |
| | | Youth and boys | PCA, Component 14 | age, boy, student, young, and year |
| | | Joy[a] | PCA, Component 4 | laugh, smile, make, run, and back |
| 3 | Domain associations | Arts and letters | Keywords | language, writer, art, artist, music, musical, poet, speech, write, and tongue |
| | | Medicine | Keywords | healer, surgeon, and heal |
| | | Sports | Keywords | athlete |
| | | Parents and art | PCA, Component 19 | mother, father, leave, art, and learn |
| | | Written communication | PCA, Component 10 | write, class, time, read, special, and student |
| | | Verbal communication | PCA, Components 20 and 2 | talk, word, people, speak, tell, understand, story, true, ask, and inside |
| | | Music | PCA, Component 7 | play, music, love, and learn |
| 4 | Beauty | Beauty | Keywords | beauty and beautiful |

The grouping of proposed themes (themselves summarized in Tables 4 and 3) into four areas reflects the authors' interpretation of the findings to synthesize the findings from the keyword investigation and the PCA.

[a] For the subsumed topic of 'Joy,' evidence of discriminant validity was inconclusive.

**Table 8. Tests for normality and equality of variances (homoscedasticity) for the prevalence of positive and negative emotions.**

| Sentiment | Normality | | | | Equality of Variances | |
|---|---|---|---|---|---|---|
| | *Gift** corpus | | *Only* corpus | | | |
| | D | p | D | p | F | p |
| Positive emotions | 0.111 | < .001 | 0.187 | < .001 | 233.674 | < .001 |
| Negative emotions | 0.165 | < .001 | 0.150 | < .001 | 2.11 | .147 |

Normality was assessed with the Kolmogorov–Smirnov Test with the Lilliefors Correction. Equality of variances (homoscedasticity) between groups was assessed with Levene's Test; for all group comparisons therein, degrees of freedom (df) was 1, with an error term 148,618.

**Table 9. Means comparisons for the prevalence of positive and negative emotions in the *gift*\* and *only* samples.**

| Emotion | Mean | | Welch | | | | Mann–Whitney | | Cohen's *d* | Cliff's *d* |
|---|---|---|---|---|---|---|---|---|---|---|
| | *gift*\* | *only* | *t* | DF | *p* | CI | U | *p* | | |
| Positive | 0.126 | 0.077 | 21.882 | 2,142.001 | < .001 | 0.045–0.054 | 199,992,159.500 | < .001 | 0.595 | 0.298 |
| Negative | 0.092 | 0.099 | -3.251 | 2,167.849 | .001 | -0.010–-0.003 | 148,126,170.500 | .002 | -0.069 | -0.039 |

## Discussion

For more than 80 years, researchers have noted a link between the broadest cultural framing of giftedness and how its manifestations are addressed within formal education [23, 106]. Yet, with few exceptions [22, 34], scientific investigations have focused on how educational stakeholders view giftedness. We termed the outlooks related to giftedness as they are found within the cultural substrate as the *cultural framing of giftedness* to distinguish our approach from earlier research in two respects. First, we considered professional cultural products [12] rather than individuals' survey responses or their impromptu actions and reactions within informal cultural products (e.g., social media). Second, we considered a context, US fictional texts (including narrative works and plays), that is removed from the perspectives of educational stakeholders, as giftedness is not first and foremost an educational concept but a widespread concept that manifests itself in thinking in a variety of situations. By way of the involved genres' relatively open remits, fictional works are understood to offer a wealth of insights into the cultures in which and for which they are created [67].

While some work has been done on investigating the meanings of giftedness in professional cultural products [13, 49–56, 58–60, 62, 63], such work has constituted cursory descriptions of a small number of cultural artifacts in most cases and has focused on educational contexts. Among the few investigations considering larger samples of professional cultural products not situated within education [58], no investigation of professional cultural products has employed methods allowing for a systematic, replicable exploration of large numbers of professional cultural products.

We focused our cultural-products investigation of the cultural framing of giftedness on a large sample of fictional texts (24,887 texts with 108,225,089 word tokens) published in the United States in recent decades (1990–2017), focusing therein on 2,104 brief passages occurring in the vicinity of the word *gift*\*, when this word was used to refer to giftedness [78]. With data representative of recent US fictional texts, we investigated two perennial research questions related to giftedness that, however, have mostly been addressed using data from educational stakeholders: Which notions of giftedness and the gifted are salient in literary texts published in recent decades in the United States? What is the overall sentiment valence when the terms *giftedness/gifted* are used in these texts?

### What is giftedness about in US fictional texts?

Our findings on the notions of giftedness and the gifted in US fictional texts (RQ 1) suggest that four broad concerns are being linked with giftedness in that professional cultural product: (a) ability, learning and education, and the rarity of giftedness, (b) causes and consequences of giftedness, (c) a specific set of domain associations, and (d) the aesthetic quality of beauty. Each area of concerns has different implications for the overall understanding of giftedness as a cultural concept.

The first area—ability, learning and education, and the rarity of giftedness—dovetails with findings on education stakeholders' associations with giftedness and the gifted. Investigations into education stakeholders' associations with giftedness and the gifted are broadly in

agreement about the term connoting domain-general intellectual prowess, precocity, or promise [21–23, 107–110]. In this area, our findings regarding the use of the term *gift\**, when used in the dictionary sense associated with giftedness, corroborate findings based on the investigation of individuals' views and of cultural products in educational settings. Here, there is consensus about the notion of giftedness between our findings and education-situated individual and artifactual investigations.

The second area—causes and consequences of giftedness—reflects cultural presuppositions about the etiology and manifestations of giftedness. In this area, our findings are also in line with research on how educational stakeholders think about the rarity or remarkability of manifestations of giftedness [108, 109, 111] as well as about the positive and negative socioemotional concomitances of giftedness, which have been characterized as harmony and disharmony lay hypotheses of giftedness [22, 112, 113]. Moreover, the associations we noted with youth and boys corroborate earlier work on giftedness suggesting that individuals may more readily ascribe intellectual giftedness or high ability to males [107, 114–116]; higher rates of participation in gifted education programs for boys despite gender-equal rates of formal gifted identification for girls and boys may also suggest this bias [117].

For the third area—a specific set of domain associations—giftedness as it is used in US fictional texts suggests an as-yet unidentified set of field-specific ability beliefs [118] being associated with the giftedness concept, which consists of words related to arts and letters, music, athletics, written and verbal communication, and medicine. The understanding of giftedness reported for investigations of educational stakeholders and education-related cultural products tends to reflect a general concept of intellectual giftedness with a varying accompanying set of psychosocial ascriptions [21, 23, 119–123] and does not provide comparable evidence of a set of domains being associated with giftedness. When studies have considered domain-specific manifestations of giftedness, they have pre-defined a selection of domains [9, 124], which is problematic in that educators have been shown to be willing to indiscriminately endorse a wide array of proposed giftedness definitions [23, 25] and to inaccurately self-assess their knowledge of intellectual giftedness [23]. Moreover, the set of fields being associated with giftedness in US fictional texts is remarkable for what it omits, namely, fields within science, technology, engineering, and mathematics (STEM)—with medicine being a possible exception—and thus roughly half of the canon of widespread learning and talent domains across primary, secondary, and tertiary education [125–130]. A better understanding of the domains commonly associated with giftedness outside of educational contexts may be useful for research on relationships between field-specific ability beliefs, social address variables (e.g., gender), and academic participation within education [118, 131, 132] by providing a better picture of the established cultural associations between a common term related to ability beliefs—*gifted*—and specific learning domains.

The fourth area of findings—the aesthetic quality of beauty—does not obviously map onto findings on perceptions of giftedness of recent decades. However, the association does recall the eugenics-inspired attempt of Francis Galton [133, 134] to create a "'Beauty-Map' of the British Isles" [133] (p.315), a mapping of his perceived physical attractiveness of women. While Galton's premise is pseudoscientific [134] and research on giftedness no longer discusses connections between giftedness and physical appearance, it is important to note that this association appears to live on in how the word is used within US fictional texts. An awareness of the giftedness–beauty association is important for research-based discussions of the proscriptive formal use of the terms *giftedness*, *gifted*, and *gift*, as labeling decisions in formal contexts should reflect a broader descriptive understanding of how the term is used in everyday contexts [14].

In sum, with respect to RQ 1, the exploratory findings reported for notions of the term *gift*\* when used in US fictional texts of recent decades in senses related to giftedness appear to function as a catch-all for a host of associations, including impressive or rare abilities, skills, and aptitudes in general and with respect to certain domains, for the positive and negative causes and consequences of giftedness, and for beauty. As such, earlier explorations of respondents' notions related to giftedness, gifted education, or gifted children appear understandably fraught [1, 19, 23, 25, 135, 136], and the situation is comparably fraught for formal definitions [6, 137–144]. The term is polysemous to an extent that is likely to have generally hindered attempts to use the term to more specific educational ends. The variety of connections observed for the term as well as its salient association with beauty suggests that it is not well-suited as a designator for more specific forms of expert-defined giftedness in formal educational contexts. This finding supports observations about the pitfalls of the term's polysemy [7, 25, 139, 145–147] and speaks in favor of suggestions about limiting or discontinuing use of the term *gifted* as a formal nomenclature within educational policy and practice [143, 146, 148–152].

## What is the affective valance of giftedness discourse in US fictional texts?

Our sentiment analysis of giftedness-related passages in US fictional texts (RQ 2), that is, in a professional cultural product that is not closely associated with education stakeholders and settings, suggests that the term is often connoting positivity. This does not imply an invalidation of earlier findings indicating ambivalent or negative appraisals of the concept [1, 15, 22, 23, 34], but requires a careful explanation. The positivity of giftedness-related passages observed in our sample of recent US fictional texts likely reflects the avoidance of educational settings and educational stakeholders and of the concomitant activation of social comparison processes in such settings that induce negative reactions.

In educational contexts, respondents often manifest negative reactions towards giftedness when asked to reflect on generic statements or short vignettes related to giftedness or the gifted [8, 22, 44, 47, 107, 153–155]. Gifted education and education generally are accompanied by a widespread tension between educational efforts to support excellence and efforts to support equity [16, 24, 143, 156, 157] as well as hostility towards intellectual elites [4, 44, 158–163] and discomfort with intellectual precocity [17, 44, 164].

Various rationales have been proposed for the negative reactions to giftedness observed in educational contexts. In an earlier study [162], the authors described how use of common gifted labels in schools fulfills the eliciting conditions of envy [165]—including a self-diminishing social comparison with others, high self-relevance of the point of comparison, and high similarity between persons involved in the comparison—and makes envious reactions to gifted labels likely among pupils. More recently [22], a rationale was articulated for negative associations with giftedness based on the stereotype content model [166], which conceptualizes individuals' ad hoc reactions to a target concept (often an elicited stereotype) as an impromptu judgment call balancing the individual's assessment of the target concept's warmth (i.e., intent to help or harm) and competence (i.e., ability to help or harm). While the more recent study [22] was designed for a nationally representative population rather than a scholastic population, one of five items tapping respondents' associations surrounding gifted individuals focused respondents' attention on grades and education, which seems likely to have created an education-related framing for respondents. Both studies [22, 34] reported some evidence of negative reactions to their giftedness targets in line with the authors' rationales. When individuals respond to generic questions about giftedness, the gifted, or gifted education, more negative associations seem plausible, then, if such assessments are taking place in contexts involving the sharing of educational resources.

In the case of our findings, the predominately positive affective valance may be reflecting the avoidance of educational settings in which evocations of giftedness elicit competitive self-comparison processes. It has been suggested that defensiveness about giftedness may disappear when the common understanding of giftedness as general intellectual or academic superiority or precocity is replaced by narrowly defined domain-specific types of giftedness or talent that do not align with the interests of the respondents [44, 140, 156, 162]. In the case of fictional texts as a professional cultural product, both the respondent–target concept dynamic and the educational setting are being avoided.

Moreover, there is good reason to assume that positive associations with giftedness generally do exist. Within research on perceptions of giftedness in educational settings, a substantial body of research indicates that respondents' attitudes towards giftedness, the gifted, and gifted education are positively correlated with the amount of exposure individuals have had to gifted education and the amount of training and on-the-job experience that teachers have [6, 20, 23, 24, 34, 123, 154, 167–173]. The latter finding suggests that systematic deconstruction of individuals' giftedness-related stereotypes by way of increased knowledge about and experience with giftedness [119] allows individuals to access and express underlying positive associations with giftedness. In other words, positive associations with giftedness exist, even in educational settings, but are being masked by the negative reactions that induced reflection on giftedness concepts tends to elicit when such concepts are activated in educational settings and respondents lack sufficient knowledge about giftedness and gifted education.

Fictional works, which are known for breadth and complexity [174], may be reflecting more positive underlying cultural perspectives on giftedness related to the upsides of giftedness as such, including the wonderment and marvel that accompany the observation of and reflection about manifestations of giftedness [175]. The observation of a predominant tone of positivity when giftedness is addressed in fictional texts provides a useful qualification for the negativity associated with giftedness in educational contexts. Negative reactions to giftedness —likely being induced by the methods and contexts with which perceptions of giftedness have traditionally been investigated, as we have argued—may be masking or overshadowing simultaneous widespread cultural appreciation of giftedness we observed in our US fictional texts sample.

By avoiding social comparison processes and educational settings, our study sheds further light on the underlying cultural framing of giftedness and provides additional support for the existence, at a cultural level, of a general tone of positivity surrounding giftedness. Our sentiment-analysis finding suggests that in nuanced literary communication that is removed from educational or other professional settings giftedness is a term that may often be invoked with a sense of wonder, joy, hope, or appreciation. This interpretation does not provide a counterargument against the idea of retiring *gifted* from its use in formal contexts noted above. Yet the affective valance surrounding giftedness as a cultural concept remains an important research question, because efforts to update or remove the usage of the term in formal contexts should be supported by descriptive information about the term's usage and cachet in cultures and subcultures in general, not just in the classroom. As practitioners, researchers, and policymakers steer parents and children away from usage of *gifted* and, for example, towards person-first language [149], such efforts will likely be more effective if a more accurate and nuanced understanding of the de facto cultural framing of giftedness is available. Such an understanding can help to bridge the gap between the everyday cultural, literary, and linguistic reality of non-specialists and the oftentimes rarified lexicon of research, education, and policymaking.

## Limitations and implications for future research

We note five limitations that should be kept in mind when considering the findings presented above. For each limitation, we provide suggestions for future research.

First, we conceived of our study as an exploratory study relying on methods of quantitative text analysis. This approach allowed us to consider a very large sample but limited our ability to look at nuance and idiosyncrasies within our data set. Our distant-reading approach [68, 80] represents a first step in examining large sets of unstructured text data and provides replicable results on which others can build. Our findings reflect a systematic method but raise new questions in need of closer scrutiny. For example, more detailed investigations of the 2,104 giftedness-related passages identified in our sample taken form the Corpus of Contemporary American English (COCA) [74] would likely allow for additional qualification of our findings; and analogous investigations of other large collections of fictional texts or other professional cultural products would allow for a replication of our methods that was not possible within this study.

Second, while the COCA was created on the basis of a systematic approach to capturing language use within a given nation and over a specific time period, it is important to note that the corpus has more in common with a convenience sample in that the inclusion of texts in the corpus was determined in part by the texts' accessibility for inclusion in the corpus [70, 72, 73]. The literature describing the creation of the COCA does not explain how the noted components making up the fiction subcorpus—"short stories and plays from literary magazines, children's magazines, popular magazines, first chapters of first edition books [. . .], and fan fiction" [73]—provide a plausible representative approximation of the entirety of contemporary fictional writing in the United States. However, in matters of literary and artistic genres, an exhaustive definition of a given genre and its component parts does not seem realistic considering the non-exact nature of artistic genres such as fiction [174]. Confidence about the appropriateness generalizability of the sample will require additional investigations of different large samples of fictional texts published in the United States.

Third, regarding the notions of giftedness and the gifted in US fictional texts (RQ 1), we noted the lopsidedness of the fields associated with giftedness. It appears in our sample that giftedness occurred with a specific set of domains that largely excluded references to STEM domains. This result could be an artifact of a selective focus on certain domains in US fictional texts. Our exploratory study design did not investigate the overall visibility or lack thereof of domains within our corpus. However, even if references to STEM domains are occurring at lower rates than, for example, humanities domains in our US fiction subcorpus overall, the keyword identification method we employed, which is based on a comparison within the same corpus, would have accounted for such a difference. In other words, if references to a given STEM domain are less common than references to a given humanities domain in our US fiction subcorpus overall, the STEM domain would nevertheless show up as a salient keyword in the context of *gift** if it were more likely to have occurred in the vicinity of *gift** than in the vicinity of our comparison word, *only*, in the same corpus, unless words relevant for recognizing certain STEM domains were totally lacking in our comparison corpus. Considering the size of our comparison corpus ($n = 146,516$ passages), a total lack of words relevant for STEM domains (e.g., words related to mathematics) is unlikely. Future research should address this question systematically when further investigating the specific set of domain associations we observed near *gift** in our COCA fiction subcorpus.

Fourth, the more generalized themes we identified regarding the notions of giftedness and the gifted in US fictional texts (RQ 1) partially reflected our own synthetic interpretation of the specific findings from our keyword identification process and our principal component

analysis. We pointed this out when describing and discussing our results. It is important that researchers distinguish between our primary findings and our explanations of these exploratory findings, in which we suggested further groupings of the numerous individual findings. Future research can meaningfully build on our findings by, for example, investigating the existence of the topics and themes we described in additional cultural products or via surveys of respondents.

Finally, regarding our sentiment analysis of giftedness-related passages in US fictional texts (RQ 2), we considered the overall balance of positive and negative sentiment words in our *gift** passages in comparison to our comparison corpus, consisting of passages centered on the grammar word *only*. Our assessment of an overall tilt towards positivity applies for the *gift** passages in our sample of US fictional texts as a whole and reflects an exploratory first step at characterizing the emotional valance of references to giftedness in fictional texts. Future work should look more closely at how the sentiment within *gift** passages varies according to various topics or themes (e.g., according to domain or social address variables) or between types of cultural product (e.g., news versus literature).

## Concluding remark

Giftedness is a polysemous lay concept rooted in culture as well as a construct used in formal education. Scientific work on perceptions of giftedness has focused on the latter context—formal education—and relied on reactive assessment methods, mostly questionnaire-based surveys of respondents. It is thus unsurprising that much of the rich cultural experience surrounding the giftedness concept is masked or filtered by the education context and its stakeholders' perspectives. To shed more light on the cultural framing of giftedness, we sought to uncover the meanings and emotional valance occurring near the term *gift**, when used in the dictionary sense of giftedness, in a large sample of recent US fictional texts. Our findings suggest a cultural framing of giftedness that differs from the concept of giftedness reported for educational contexts. Invocations of giftedness in recent US fictional texts include domain-specific associations within arts and humanities (but not in STEM), a connection with beauty, and, on balance, a positive emotional stance. By complementing the findings of earlier scientific work, the findings contribute to a more nuanced understanding of cultural conceptions of giftedness. This is particularly useful at present, as scholars are exploring options for moving away from the *gifted* nomenclature in formal education on account of the term's connotations and implications. A better understanding of term's broader cultural framing will allow for a more thorough review of its meanings and support more data-driven decisions about its use or avoidance in educational settings.

## Supporting information

**S1 File. Technical supplement to cultural framing of giftedness in recent US fictional texts.** Contains all instructions and code needed to replicate our study.
(PDF)

## Acknowledgments

We thank the Open Access Team of the University of Regensburg University Library for their assistance in setting up a long-term repository for our study data.

## Author Contributions

**Conceptualization:** Daniel Patrick Balestrini, Heidrun Stoeger.

**Data curation:** Daniel Patrick Balestrini.

**Formal analysis:** Daniel Patrick Balestrini.

**Investigation:** Daniel Patrick Balestrini.

**Methodology:** Daniel Patrick Balestrini, Heidrun Stoeger.

**Resources:** Daniel Patrick Balestrini.

**Software:** Daniel Patrick Balestrini.

**Validation:** Daniel Patrick Balestrini.

**Visualization:** Daniel Patrick Balestrini.

**Writing – original draft:** Daniel Patrick Balestrini.

**Writing – review & editing:** Daniel Patrick Balestrini, Heidrun Stoeger.

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
