## [Decision Letter · Decision Letter 0]

15 May 2024

PONE-D-24-06759Cultural framing of giftedness in recent US fictionPLOS ONE

Dear Dr. Balestrini,

Thank you for submitting your manuscript to PLOS ONE. After careful consideration, we feel that it has merit but does not fully meet PLOS ONE’s publication criteria as it currently stands. Therefore, we invite you to submit a revised version of the manuscript that addresses the points raised during the review process.

**Thank you for submitting this material. Based on the reviewer's feedback, we need to make the following revisions:**

**We need to further elaborate the perspective and provide more supporting evidence**. Please add relevant examples and data to strengthen your argument.

In the conclusion section, we need to summarize the key points of the article more clearly and concisely. Try to encapsulate the main thesis and core conclusions in a more succinct manner.

Overall, please carefully check the logic and coherence of the article to ensure smooth transitions between paragraphs. Also pay attention to the accuracy and literary quality of the phrasing.

Please make the revisions based on this feedback and resubmit the updated version. I will carefully review your changes and provide further suggestions if needed. Feel free to reach out if you have any questions.

We look forward to receiving your revised manuscript.

Kind regards,

Luobing Dong

Academic Editor

PLOS ONE

Journal Requirements:

Additional Editor Comments:

This paper have conducted a thorough investigation into the portrayal of giftedness in recent US fiction using quantitative text analysis. Its findings suggest that the term "gifted" is often associated with humanities and beauty in novels and has a generally positive emotional valence. However, the study's limitations should be considered, such as potential biases in sample selection and the focus solely on novels as a cultural product. Despite these limitations, the study provides valuable insights into how American society views giftedness. Overall, the authors' approach and methodology are sound, and their findings contribute significantly to the ongoing discourse on giftedness.

Reviewers' comments: 

Reviewer's Responses to Questions

**Comments to the Author**

1. Is the manuscript technically sound, and do the data support the conclusions?

Reviewer #1: Yes

2. Has the statistical analysis been performed appropriately and rigorously? 

Reviewer #1: Yes

3. Have the authors made all data underlying the findings in their manuscript fully available?

Reviewer #1: Yes

4. Is the manuscript presented in an intelligible fashion and written in standard English?

Reviewer #1: Yes

5. Review Comments to the Author

**Reviewer #1:** Cultural framing of giftedness in recent US fiction

This paper investigated the cultural meanings associated with ‘gifted’ and/or ‘giftedness’ through the analysis of US fiction corpus. The authors have endeavoured to review the terms apart from the ordinary angle of perceptions and attitudes surveys usually concerning the context of formal education. The Corpus of Contemporary American English (COCA) was used as a major source for the recent decades of US fiction. Researchers aims to locate the notions of the highlighted terms by examining the salient words types (i.e., themes) used before and after the terms and the sentiment when these expressions are used in texts. Outcomes from the keyword identification procedure shows that in general those terms are located mostly with positive meanings. The obtained meanings were mainly around ten possible themes such as, potent natural ability and written communication. For the sentiment analysis the authors used a lookup dictionary for positive and negative emotions to examine the extract corpus. Results indicate a significant difference for the positive emotion words comparing to the negative ones.

Overall I find the study well designed and executed and the paper well-presented and thorough. I find the research questions to be answered appropriately by the method and the analysis used. Nonetheless, I am a little more skeptical than the authors regarding the manual coding process to remove unrelated usage of ‘giftedness’, the possibility of errors being included cannot be ruled out. Therefore, it may be more reliable to use an electronic instrument for the analysis or at least use another reviewing eye for their manual coding procedure. The same would be applicable for the process of resolving the encoding errors. Moreover, authors need to be clear with regard to the tool used to locate the found results for instance, when limiting their keyword investigation to word tokens occurred in target and reference corpus. Also, when they built a matrix for target and reference corpus, is there any software assisting in the step? This should help in study reduplication in a different context and clarifying the process followed for general readers and unexperienced researchers. In addition, I certainly understand the authors attempts to advertise their gap knowledge as cultural products are essential in indicating the views on gifted people (i.e., giftedness), however, the educational setting surveys are to complete the picture and though shouldn’t appear in contracting place in the paper.

Ultimately, I think the paper has a great potential to establish several more important bridges on using corpus to investigate a nation views regarding specific term. The study is interesting and generally well though through, and there is certainly merit in examining if general minor corrections mentioned above and the ones to be followed were addressed.

Minor issues

Line 125: try to break your literature into more titles for instance, line 125 should be something like ‘studies on cultural products on/of giftedness’

Line 196: can be inserted on your line 220 under the present research with doing necessary corrections/amendments.

Line 321: in the sentence ‘In unclear cases’, it is unclear what do you mean (rephrase) the sentence.

Lines 331-332: have you resolved encoding errors manually? Clarify.

Line 354: better write it as ‘we will describe the implemented analytical procedures’.

Line 355: it would be better to refer to research questions one or two by using RQ1 or RQ2 starting from its second appearance on text.

Lines 354-355 and 359-365: you have used the word ‘implement’ a lot try to find another synonymous.

Line 386: what the tool you used to limit your keyword investigation?

Lines 389-390: is there a software assisting you in built matrix, if so please specify.

Line 390: what (2-) is used for? You mean at the two columns?

Line 393: you mentioned “we then assess using an inferential statistical measure”, what is that? Again, have you completed through a software? Always draw the complete picture for your readers.

Lines 400-401: what is the Principal Component Analysis (PCA)? Who invented it and why it would be reliable in such a case?

Line 402: “was not needed” why? remind the reader of your aim.

Line 414: at the end of this sentence, it seems would need to continue speaking about second or third followed procedure. Therefore, whether you delete (first) from the sentence, so the reader wouldn’t expect the continuity of ideas or add next processe/s to the same paragraph.

Line 424: what are the bases for your cutoff point? You seem to provide an answer at line 430 so try to relocate the sentence.

Lines 432-436, try to break the sentence for better comprehension.

Line 437: “The document-feature matrix consisted of 187,406 values of 0”, you mean at 187,406 text instances you have got 0 word type? It doesn’t make sense as you have found more with lower text instances, please defend/explain your finding.

Line 438-439: try to differentiate between text instances and values found with paratheses for instance, 5.813 (1 value) or any other style.

Line 441: write the numbers/equation in a better way.

Line 464: how you entered the target words in the dictionary, manually?

Lines 566-568: unclear sentence, rewrite please.

Lines 610-611: rewrite the sentence to reduce the use of ‘for’ in it.

Lines 614-617: you have already mentioned the components above so, no need for the repetition in these lines.

Lines 697-698: “it is reasonable in the context of exploratory research”, according to what you have built this conclusion?

Line 784: “to live on in” ?

Line 843: replace ‘within’ with ‘in’ to avoid repetition.

Line 856: “as such, such as” replace one.

Lines 869-873: a prolonged sentence, try to break it up.

6. PLOS authors have the option to publish the peer review history of their article (what does this mean?). If published, this will include your full peer review and any attached files.

Reviewer #1: No

---

## [Author Response · Author response to Decision Letter 0]

25 Jun 2024

We submitted a properly formatted version of our responses as a PDF file. Here, I'm copying those responses into this field.

Comments by the Academic Editor

Thank you for submitting this material. Based on the reviewer's feedback, we need to make the following revisions:

Thank you for your comments as academic editor! We attended to all comments.

We need to further elaborate the perspective and provide more supporting evidence. Please add relevant examples and data to strengthen your argument.

We attended to all of the instances in which the reviewer requested more elaboration/explanation, and this involved adding an additional source of supporting data to our manuscript. For example, we …

• Elaborated the relationship between our work and earlier work by adding additional references to how our approach complements earlier work in the Introduction (see Lines 109–110 in the change-tracked revised manuscript) and in the Concluding Remark (see Lines 1,044–1,045);

• Added an external inter-rater agreement test after our manual coding procedure (to show that our coding procedure was reliable);

• Revised how we deal with cleaning of the original corpus data to be more systematic;

• Resolved two issues raised by the reviewer regarding our methods (tools, building of our matrix) by adding additional references to our Technical Supplement (S1 File); and

• Improved our documentation of the procedure we used to identify a word-inclusion threshold for our document–feature matrix.

You will find longer descriptions of these revisions among our responses below.

In the conclusion section, we need to summarize the key points of the article more clearly and concisely. Try to encapsulate the main thesis and core conclusions in a more succinct manner.

We revised our conclusion section (Concluding Remark) to summarize the key points of the article more clearly and succinctly.

Overall, please carefully check the logic and coherence of the article to ensure smooth transitions between paragraphs. Also pay attention to the accuracy and literary quality of the phrasing.

We reread the entire manuscript to check its logic and coherence and to ensure smooth transitions between paragraphs. We made numerous small changes to improve clarity, reading flow, and the accuracy and consistency of wordings and phrasing. For example, we standardized our use of the terms ‘fiction’ and ‘fictional texts’ with ‘fictional texts’ throughout the entire manuscript.

A native speaker of English read the manuscript and gave us feedback on phrasing and idiomatic style. This helped us to root out typos, revise awkward turns of phrase, and avoid wording inconsistencies.

Journal Requirements:

Thank you for directing our attention to these resources. We reviewed both documents and revised the headings, tables, figures, bibliography, and file names in accordance with the PLOS ONE standards. We processed our figure images with the PACE image-editing tool to ensure they comply with PLOS ONE image guidelines.

We reviewed the reference list and made formatting adjustments to comply with the PLOS ONE (Vancouver) reference system. In one case, we had accidentally referenced the same publication with two different numbers. We removed the duplicate reference. We added one new reference in response to reviewer feedback. The added reference is noted in our response to the reviewer (below), and both the removed duplicate reference and the added reference are pointed out in the change-tracked revised manuscript. Finally, we realized that we had included one parenthetical literature citation in the manuscript. We provided a proper PLOS-ONE style reference for this citation.

Additional Editor Comments:

This paper investigated the cultural meanings associated with ‘gifted’ and/or ‘giftedness’ through the analysis of US fiction corpus. The authors have endeavoured to review the terms apart from the ordinary angle of perceptions and attitudes surveys usually concerning the context of formal education. The Corpus of Contemporary American English (COCA) was used as a major source for the recent decades of US fiction. Researchers aims to locate the notions of the highlighted terms by examining the salient words types (i.e., themes) used before and after the terms and the sentiment when these expressions are used in texts. Outcomes from the keyword identification procedure shows that in general those terms are located mostly with positive meanings. The obtained meanings were mainly around ten possible themes such as, potent natural ability and written communication. For the sentiment analysis the authors used a lookup dictionary for positive and negative emotions to examine the extract corpus. Results indicate a significant difference for the positive emotion words comparing to the negative ones.

Overall I find the study well designed and executed and the paper well-presented and thorough. I find the research questions to be answered appropriately by the method and the analysis used. 

Thank you for your positive feedback and your review of our manuscript! We appreciate your comments, questions, and suggestions and have revised our manuscript to address all of the concerns raised by the academic editor and the reviewer. We respond to all points below, point by point.

Nonetheless, I am a little more skeptical than the authors regarding the manual coding process to remove unrelated usage of ‘giftedness’, the possibility of errors being included cannot be ruled out. Therefore, it may be more reliable to use an electronic instrument for the analysis or at least use another reviewing eye for their manual coding procedure.

Thank you for pointing this out!

To address this concern, we recruited an independent rater who independently coded a systematic sample of 200 of the 9,151 passages we manually coded. The 200 passages we used for the external coding were spread out evenly over the 9,151 passages. The rater worked independently following the procedure we originally used (and described in the original manuscript and the Technical Supplement). The independent-rater test of our coding procedure showed that our manual coding procedure was highly accurate, with an agreement of 93%. We added information about this to the manuscript and details of the coding procedure to the revised Technical Supplement. See Lines 336–344 in the change-tracked revised manuscript and pp. 29–32 in the Technical Supplement. We have also included the instruction letter we sent to the manual coder and the file the coder returned in the revised version of our data repository. We added one additional reference to the literature providing a basis for our external-rating procedure (1), which is listed as Reference Number 78 in the change-tracked revised manuscript (see Lines 1,275–1,277).

The same would be applicable for the process of resolving the encoding errors. 

Thank you for pointing this out!

We developed and implemented a more systematic approach to dealing with encoding issues. We revised the first part of our data-cleaning procedure. See Lines 347–362 in the change-tracked revised manuscript and Section 7.2 (pp. 43–53) in the revised Technical Supplement.

Because the new procedure changed the corpus on which we based our analyses, we reran the entire code pipeline of the study. This led to small changes in the values we report for the analyses. All such changes are indicated in the change-tracked revised manuscript.

Moreover, authors need to be clear with regard to the tool used to locate the found results for instance, when limiting their keyword investigation to word tokens occurred in target and reference corpus.

For all steps, we used existing tools (code packages) running in R and, in a few cases, in Python as well as custom-built functions. For example, a key package of tools for text mining are the functions provided by the Quanteda suite of text-mining tools. The Technical Supplement includes all the code and tools we used to carry out the study. It explains how to use the tools and provides comments for the custom functions we wrote. Readers should be able to replicate our study by following the Technical Supplement and employing the same tools described therein. The supplement also includes version information and citations with bibliographic references (at the back of the supplement) for all the existing text-mining tools we employed.

In the changed-tracked revised manuscript, we added two additional remarks referring readers to the Technical Supplement to help readers find the information on our tools and methods. In the changed-tracked revised manuscript, we added a brief reference to the supplement at Lines 279–280 and a longer explanation of the supplement in Lines 376–379.

Also, when they built a matrix for target and reference corpus, is there any software assisting in the step? This should help in study reduplication in a different context and clarifying the process followed for general readers and unexperienced researchers.

We explain the matrix creation in detail and provide a bibliographic reference for the text-mining function we used (the Quanteda dfm() function) in the supplement. See the first sentence in Section 9 and Listing 9.1 on p. 71 in the revised Technical Supplement.

We added an extra remark to the main manuscript when we mention creating the matrix for the keyword identification procedure. The new remark directs readers to the supplement regarding the creation of the matrix. See the new parenthetical remark in the revised changed-tracked version of the main manuscript at Lines 424–425.

We understand that not providing the names of such functions and tools in the main manuscript means the reader of the main manuscript will have to cross-check technical details about functions and the like in the supplement. However, we decided it was a better trade-off for the readability of our main manuscript.

In addition, I certainly understand the authors attempts to advertise their gap knowledge as cultural products are essential in indicating the views on gifted people (i.e., giftedness), however, the educational setting surveys are to complete the picture and though shouldn’t appear in contracting place in the paper.

Thank you for this comment.

To make clear that we view our approach not as based a contradictory claim but as one offering an additional perspective, we frame the investigation of cultural products as a “complementary method” (see Line 105 in the change-tracked revised manuscript). Then, in the Discussion, we describe how our approach and findings elaborate or complement the picture created by earlier educational-settings surveys. In Lines 890–892 of the changed-tracked revised manuscript, we explain that our findings do not invalidate earlier findings based on surveys done in educational settings, but rather provide a complementary perspective: “This does not imply an invalidation of earlier findings indicating ambivalent or negative appraisals of the concept […] but requires a careful explanation.” After that remark, we then provide an explanation of how our findings build on the earlier findings (rather than invalidating them).

To address your concern, we added additional remarks drawing readers’ attention to the complementary nature of our approach and findings:

• In the Introduction, we added an additional sentence in Lines 109–110 (in the change-tracked revised version) to elaborate how our work builds on earlier findings by providing additional context.

• In the Concluding Remark, we added an additional adverbial phrase to remind readers at the very end of the manuscript that we view our research as a complementary approach providing additional insights complementing the earlier work: “To shed more light on the cultural framing of giftedness,” (see Line 1,037–1,038 in the changed-tracked revised manuscript).

Ultimately, I think the paper has a great potential to establish several more important bridges on using corpus to investigate a nation views regarding specific term. The study is interesting and generally well though through, and there is certainly merit in examining if general minor corrections mentioned above and the ones to be followed were addressed.

Thank you very much for your positive assessment of our work and your helpful feedback on our manuscript. We appreciate all the points you have made and have endeavored to improve the manuscript by addressing all of them.

Minor issues

Line 125: try to break your literature into more titles for instance, line 125 should be something like ‘studies on cultural products on/of giftedness’

Thank you for this helpful comment. We added two level-3 headings to break the literature review in the section to which you refer into smaller units of information. You will find the new sub-sub-headings at Lines 117 and 129 in the revised change-tracked manuscript.

Line 196: can be inserted on your line 220 under the present research with doing necessary corrections/amendments.

Following your suggestion, we moved the leve-2 heading “The present research” one paragraph up (earlier) in the manuscript. The heading is now located on Line 203 in the revised change-tracked manuscript.

Line 321: in the sentence ‘In unclear cases’, it is unclear what do you mean (rephrase) the sentence.

We rewrote the phrase to provide a better explanation of what we mean. We deleted “In unclear cases” and instead wrote “When the 51-word text windows provided insufficient context for determining the sense of the lexeme “gift,”.” For the manual coding process, it was usually obvious which sense of the word was being used by looking at a few words to the left and right of the node word ‘gift.’ You will find the revision on Lines 333–335 of the changed-tracked revised manuscript. In some cases, more context was needed. For that reason, the manual coding process included longer excerpts with 100 words before and after ‘gift.’ We also explain this point in the Technical Supplement. See p. 28 (second paragraph from top) in the revised supplement.

Lines 331-332: have you resolved encoding errors manually? Clarify.

Yes, we used a manual process. However, in light of the earlier related point (above), we revised our procedure here. See our detailed explanation above in this letter.

Line 354: better write it as ‘we will describe the implemented analytical procedures’.

Following your suggestion, we changed the tense of the statement from simple present to future by adding a “will” in Line 387 of the revised change-tracked manuscript.

Line 355: it would be better to refer to research questions one or two by using RQ1 or RQ2 starting from its second appearance on text.

We adopted this suggestion throughout our manuscript. At the first occurrence of “Research Question 1” (Line 246 in the revised change-tracked manuscript) we introduce the abbreviation “RQ1.” At the first occurrence of “Research Question 2” (Line 249 in the revised change-tracked manuscript) we introduce the abbreviation “RQ2.” We maintained the written-out forms of both terms when “Research Question” appears in a heading, but added the abbreviation to the heading to help readers remember the abbreviation (see Lines 539 and 738).

Lines 354-355 and 359-365: you have used the word ‘implement’ a lot try to find a

---

## [Editor Report · Decision Letter 1]

2 Jul 2024

Cultural framing of giftedness in recent US fictional texts

PONE-D-24-06759R1

Dear Dr. Balestrini,

We’re pleased to inform you that your manuscript has been judged scientifically suitable for publication and will be formally accepted for publication once it meets all outstanding technical requirements.

Kind regards,

Luobing Dong

Academic Editor

PLOS ONE

Additional Editor Comments (optional):

The authors responded all comments. They checked the quality of the paper and this version is ready to accept.
---

## [Editor Report · Acceptance letter]

8 Jul 2024

PONE-D-24-06759R1 

PLOS ONE

Dear Dr. Balestrini, 

I'm pleased to inform you that your manuscript has been deemed suitable for publication in PLOS ONE. Congratulations! Your manuscript is now being handed over to our production team.

Kind regards, 

on behalf of

Dr. Luobing Dong 

Academic Editor

PLOS ONE